# Rotary Kiln, a Unit on the Border of the Process and Energy Industry—Current State and Perspectives

Jiří Bojanovský [1], Vítězslav Máša [1,*], Igor Hudák [1], Pavel Skryja [1] and Josef Hopjan [2]

1    Faculty of Mechanical Engineering, Brno University of Technology, Technicka 2, 616 69 Brno, Czech Republic
2    Trinom Inc., Tovacovská 3000/11a, 750 02 Prerov, Czech Republic
*    Correspondence: masa@fme.vutbr.cz

**Abstract:** A rotary kiln is a unique facility with widespread applications not only in the process industry, such as building-material production, but also in the energy sector. There is a lack of a more comprehensive review of this facility and its perspectives in the literature. This paper gives a semi-systematic review of current research. Main trends and solutions close to commercial applications are found and evaluated. The overlap between process and energy engineering brings the opportunity to find various uncommon applications. An example is a biogas plant digestate treatment using pyrolysis in the rotary kiln. Artificial intelligence also finds its role in rotary kiln control processes. The most significant trend within rotary kiln research is the waste-to-energy approach in terms of various waste utilization within the process industry or waste pyrolysis in terms of new alternative fuel production and material utilization. Results from this review could open new perspectives for further research, which should be focused on integrated solutions using a process approach. New, complex solutions consider both the operational (mass calculations) and the energy aspects (energy calculations) of the integration as a basis for the energy sustainability and low environmental impact of rotary kilns within industrial processes.

**Keywords:** rotary kiln; cement kiln; waste; alternative fuel; combustion; artificial intelligence



## 1. Introduction

A rotary kiln (RK) is a unique facility on the border between energy and process engineering. The typical layout of an RK is in Figure 1. It serves as an example of a valuable technological combination and a representation of process intensification. The RK is a traditional technology in lime and cement production and also in many other applications. However, RKs usually consume or produce a lot of energy in the process; therefore, the current literature considers this facility more as a process or energy unit. This review paper aims to discuss the future development of this technology in the context of increasing energy prices.

RKs were developed at the end of the 19th century. Nowadays, they are used in many industrial applications (see Figure 2). The most typical RK application is in lime and cement production. However, due to RKs' unique characteristics (high capacity, high process temperatures), there are several other applications, such as various mineral resource processing, waste incineration or pyrolysis, soil annealing, etc. The main purpose of the rotary kiln is drying, combustion, calcination, pyrolysis, or the thermochemical treatment of various materials. An RK could be heated by gas (process burners), liquid, or solid fuel. RKs are usually very energy demanding.

An RK is a cylindrical vessel that rotates around the longitudinal axis. A detailed description of its function can be found in Section 3. The advantages of RKs are the ability to process diverse feedstock and control the residence time (the tilt and rotation speed of the kiln can be adjusted). On the other hand, dust generation, unstable product quality, and low thermal efficiency are barriers to the wider utilization of RKs [1].

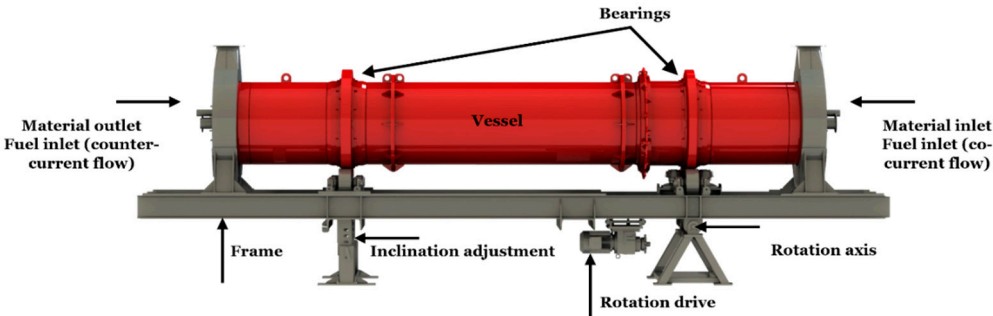

**Figure 1.** Rotary kiln layout.

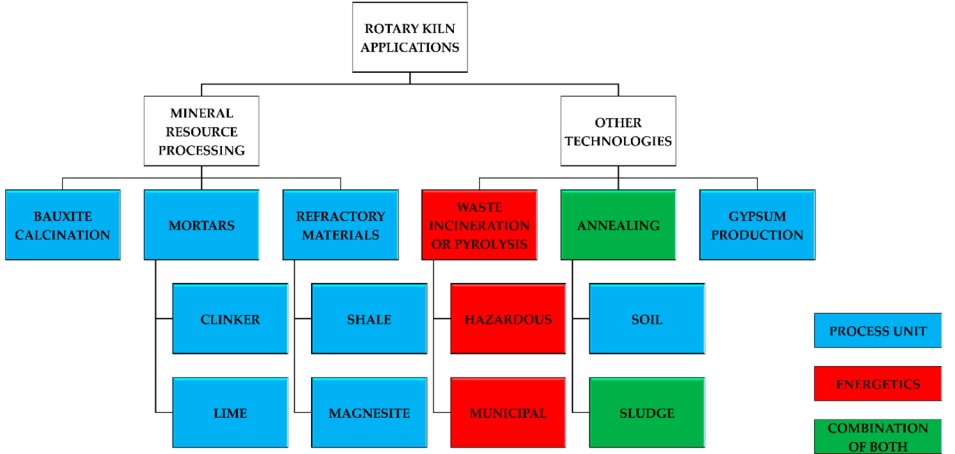

**Figure 2.** Rotary kiln industrial applications.

Cement and lime production has a significant negative environmental impact. Thus, rotary kilns are usually enhanced with preheaters and precalciners in order to increase efficiency through heat recovery. Current research is aimed at the use of alternative fuels in cement kilns (refuse-derived fuels or waste tire incineration are already used on a standard basis). The Ph.D. thesis by Nielsen [2] gives a comprehensive review of various alternative fuels combusted in cement kilns, including constructional and operational recommendations (ash from the combusted waste can cause unstable product quality, etc.). Many studies are focused on solar energy as rotary kilns can utilize the benefits of concentrated solar power. This research seems to be promising because the range of suitable applications is extensive (syngas production, chemical commodity production, ore/waste processing, etc.). However, so far, most of the published case studies have been carried out at a laboratory or pilot scale. A comprehensive review of the use of RKs for solar thermal applications is presented by Alonso et al. [3] and is further described in Section 4.2.

Nowadays, the expansion of information technology into RK control and optimization seems to be significant. There are several complex and subsequent processes in rotary kilns, the control of which is quite a complex task. Thus, artificial intelligence or numerical computational methods, such as the discrete element method, are studied in order to be used in the control systems of RKs. An excellent example of the potential of these methods is a research study by Finnie et al. [4] concerning the relation between the fill ratio and the rotational speed on longitudinal and transverse mixing (Section 4.4.).

Many reviews focus on the processes in the rotary kiln such as building material production optimization, solar energy utilization, waste or biomass material/energy utilization, etc.; others deal with control system optimization. However, a complex review of current research activities is still missing. This review paper gives a historical and theoretical background, an introduction to the current state of research, and also aims to find the connection between two different areas of application—the process industry and

the energy sector. The aim is to connect two different research perspectives to strengthen their advantages and weaken their disadvantages.

This review paper sets the following questions:

- What are the main rotary kiln research areas?
- What are the main motivations for the research on rotary kilns?
- What are the ways towards the energy sustainability of rotary kiln operation?

## 2. Methodology

This paper is written as a semi-systematic review to overview the topic by studying its progress over time or its development in crosswise research tradition. It aims to find all relevant research approaches that impact the studied topic in order to synthesize them [5]. The initial article database was created using the keywords "rotary kiln" and "cement kiln" in the ScienceDirect® platform. This database lists more than 6000 articles; only review articles were chosen for future analysis. About 400 review papers were found, but only about 150 papers focused on the research of rotary kilns. Due to their importance or added value, several research articles or Ph.D. theses were chosen to extend this database. Several groups of specializations were identified, such as clinker and lime production (Section 4.1), solar energy utilization and storage (Section 4.2), waste-to-energy systems (Section 4.3), and computational modeling, controlling, and artificial intelligence (Section 4.4); these were further divided into more specific areas such as pyrolysis and sewage sludge and waste tire treatment. The final selection of 131 studied papers was based on the originality and scope of the provided research.

In accordance with the focus of the articles, several of them were used for facility description and categorization (Section 3), while others focused on finding research trends or bringing about the answers to the scientific questions mentioned above (Section 4). The main results and findings are summarized in the tables at the beginning of every subsection of Section 4 (excluding Section 4.4.). The articles in these tables are sorted by the year of publication. Section 3 is integrated to harmonize the terminology and explain the main principle and the most important quantities. The articles in Section 4 are divided, according to their main focus, into "Energy", "Process", and "Combination". The combination of energy and process engineering is one of the main principles of the process intensification approach. The categorization can be found in the last column in the summarizing tables in Section 4. These tables also contain information on the article's importance for this paper.

## 3. Technical and Theoretical Background

The rotary kiln is a cylindrical vessel that rotates around the longitudinal axis. It can operate in a batch or continuous mode. Solid material (feedstock) enters the kiln at the higher end and leaves at the lower end. The gas passes through the kiln in the same direction as the feedstock (co-current flow) or the opposite (counter-current flow). A brief schematic layout of an RK is in Figure 1.

The longitudinal axis of the kiln can have a certain inclination (usually marked as "s" (°)) to the horizontal plane. When the axis and horizontal plane are parallel, *s* is equal to 0. Thus, due to a certain inclination, the particles move in an axial direction. The radial movement is caused by the rotation of the kiln (characterized by rotational speed $N$ ($s^{-1}$)) and the *fill ratio* (marked as *FR%* (-)). The fill ratio describes the percentage of kiln volume filled by the feedstock [3]. A typical factor connected with RK operation is the *diffusion coefficient* (marked as D ($m^2$/s)). It represents the mass of a substance diffusing through the surface in time.

The main characteristics that define the performance of rotary kilns are the type of bed motion, given by the *Froude number Fr* (-), and the residence time $\tau$ (hr) [3]. There are three main forms of transverse movement: slipping, cascading, and cataracting. In addition, seven secondary subtypes of transverse motion could be identified [6].

Bed motion diagrams were introduced by Henein et al. [7]. According to these diagrams, a certain bed motion could be identified for a certain type of particle as a function of

the Froude number (marked as *Fr*) and fill ratio (FR%). Equation (1) shows how to calculate the Froude number [7].

$$Fr = \frac{\omega^2 \cdot r}{g} = \frac{(2\pi N)^2 \cdot r}{3600 \cdot g} \tag{1}$$

According to Equation (1), the Froude number is influenced only by the angular speed $\omega$ (rads$^{-1}$), internal radius *r* (m), and gravitational acceleration *g* (ms$^{-2}$).

A centrifugal bed motion corresponds with the Froude number equal to 1 (centrifugal and gravitational forces are equal). In every other type of bed motion, the Froude number is lower than 1. The bed motion is also influenced by particle characteristics such as density, angle of repose, size, or shape [8].

When the particles lie on the bottom of the kiln, the bed motion is called the rolling mode. Due to the rotational speed, the particles roll down on the highest layer of the bed. This layer is called active, while the others are passive [8]. To improve mixing or control the residence time, the inner surface of the kiln could be enhanced with lifters, dams, fins, or paddles [9]. Another characteristic of the kiln is the residence time. It represents the time the feedstock needs to move from the input to the output. The prediction of the residence time (usually marked as τ) is quite difficult and varies with many parameters. A complex model of residence time estimation was presented by Renaud et al. [10]. Some correlations of residence time were compared with experimental data; it was found that most of the data underestimated the real residence time. Renaud et al. [10] state that one of the most accurate correlations is proposed by Sai et al. [11].

An advantage of the RK is a high operational temperature, which can, inside the RK, reach 2000 °C. Thus, the inner surface is usually made of refractory materials. The kiln can be heated up by the flame from the burner in the kiln—a direct-heated RK. In the indirect-heated RK, combustion gases or steam usually heat the outer surface of the kiln.

The indirect RK is used when the contact between the material and the heating gas is undesirable. The typical industrial application of these kilns is the processing of valuable materials in small quantities [1]. They are also used, for example, for waste remediation (pyrolysis, etc.).

Rotary kilns are mostly used in building-material production. Another common application is in waste-to-energy systems. Within combustion processes, a devolatilization occurs. This is a process where volatiles are driven out from fuel or waste (hydrocarbon-based material). Due to the high temperature, feed capacity, and continuous removal of ash content, an RK is capable of destroying toxic compounds. The RK could be equipped with a secondary combustion chamber (two-stage combustion) where the volatilized compounds from the primary chamber are incinerated [12].

Figure 2 provides a brief classification of rotary kilns according to their industrial purpose. Figure 2 was created as a combination of consultation and market research. Rotary kilns could be divided into kilns for mineral resource processing and others. Kilns for mineral resource processing are used for bauxite calcination, mortar processing (clinker and lime production), and refractory material production. The other processes that usually appear in rotary kilns are waste incineration, annealing, or gypsum production. The color divides the unit into three basic categories in agreement with the methodology of the review.

Another technology with very similar characteristics to the rotary kiln is a rotary dryer. It is an industrial facility intended to minimize the moisture of input material. The principle of the rotary dryer function is very similar to that of rotary kilns. However, due to lower process temperatures, there is usually no need to use refractory materials that require a high amount of energy for preheating, so the facility's efficiency is high. Rotary dryers are not included in this review paper.

## 4. Results—Overview of Recent Research Activities

As mentioned in Section 2, the initial database consisted of 400 review articles. A total of 131 articles (including the original research papers) connected with RK research were

chosen and cited in this review paper. The research is not focused only on common operation principles but also on the new employment and ways of RK utilization. The historical background is presented at the beginning of Section 1. The research development of the RK is presented in Figure 3 in this section. Traditional ways of RK utilization are discussed in Section 4.1 (clinker and lime production—11/131 articles were chosen and cited in this subsection) and Section 4.3 (utilization of RK in waste-to-energy systems—86/131 articles). On the other hand, Section 4.2 (solar energy utilization—16/131 articles) reveals a new approach to RK utilization within solar energy systems. Section 4.4 (computational modeling, controlling, and artificial intelligence—9/131 articles) deals with enhancing rotary kilns with computer-aided technologies.

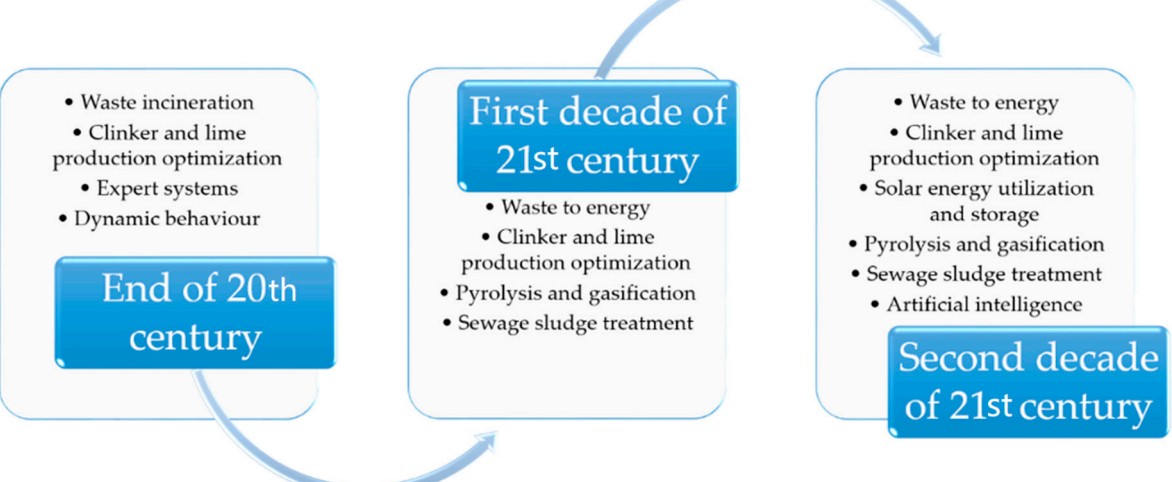

**Figure 3.** Development of main research areas for rotary kilns in the past three decades.

For the last three decades, the main areas of rotary kiln research can be traced. At the end of the last century, waste incineration and a new approach of expert systems or dynamic behavior numerical analysis were implemented. Due to the significant negative impact of clinker and lime production, its optimization could be found in the abovementioned periods. The following decade is characterized by waste-to-energy systems (including sewage sludge and waste tire treatment) and by the pyrolysis and gasification of not only waste but also biomass. The previous decade's research is similar to the first decade of the 21st century. Moreover, the solar energy utilization and artificial intelligence used within RK processes were studied. The overview of this research development is shown in Figure 3.

The summarizing tables at the beginning of every subsection in Section 4 (excluding Section 4.4) are sorted by year of publication. The articles are divided, according to their main focus, into "Energy", "Process", and "Combination". The categorization can be found in the last column, "Ca".

### 4.1. Clinker and Lime Production

Just 11 of 131 review and research articles were chosen and cited in this section, as shown in Table 1. These articles are focused on RK efficiency, heat loss reduction, fossil fuel combustion reduction, alternative fuels, etc.

**Table 1.** Articles reviewed in Section 4.1.

| Nm. | Authors | Title | Year | Motivation—Remarks | Ca. | Cit. |
|---|---|---|---|---|---|---|
| 1 | Engin T, Ari V. | Energy auditing and recovery for dry type cement rotary kiln systems—-A case study | 2005 | Optimization—RK heat loss mitigation and heat recovery in cement production | P | [13] |
| 2 | Saidur R, Hossain MS, Islam MR, Fayaz H, Mohammed HA. | A review on kiln system modeling | 2011 | Optimization—Modeling of cement kiln | P | [14] |
| 3 | Schorcht F, Kourti I, Scalet BM, Roudier S, Delgado Sancho L. | Best available techniques (BAT) reference document for the production of cement, lime and magnesium oxide: Industrial Emissions Directive 2010/75/EU (integrated pollution prevention and control) | 2013 | Optimization—Best available technologies in cement and lime production | P | [15] |
| 4 | Rahman A, Rasul MG, Khan MMK, Sharma S. | Impact of Alternative Fuels on the Cement Manufacturing Plant Performance: An Overview | 2013 | New fuels—Alternative fuels in cement production | P | [16] |
| 5 | Ishak SA, Hashim H. | Low carbon measures for cement plant—a review | 2015 | Environmental protection—$CO_2$ mitigation in cement production | P | [17] |
| 6 | Luo Q, Li P, Cai L, Zhou P, Tang D, Zhai P, Zhang Q. | A Thermoelectric Waste-Heat-Recovery System for Portland Cement Rotary Kilns | 2015 | Energy savings—Energy loss mitigation using the thermoelectric system on the RK surface | P | [18] |
| 7 | Shahin H, Hassanpour S, Saboonchi A. | Thermal energy analysis of a lime production process: Rotary kiln, preheater and cooler | 2016 | Optimization—Optimization of energy consumption in lime production | P | [19] |
| 8 | Eurostat | Energy balances (2018 edition) | 2018 | Optimization—Energy balance statistics | P | [20] |
| 9 | Rehfeldt M, Worrell E, Eichhammer W, Fleiter T. | A review of the emission reduction potential of fuel switch towards biomass and electricity in European basic materials industry until 2030 | 2020 | New fuels—Fossil fuel replacement possibility | P | [21] |
| 10 | Zaferani, SH, Jafarian, M, Vashaee, D, Ghomashchi, R. | Thermal management systems and waste heat recycling by thermoelectric generators—an overview | 2021 | Energy savings—Energy loss mitigation using the thermoelectric system on the RK surface | P | [22] |
| 11 | Antunes, M, Santos, RL, Pereira, J, Rocha, P, Horta, RB, Colaço, R. | Alternative clinker technologies for reducing carbon emissions in cement industry: A critical review | 2022 | Alternative technologies of clinker production | P | [23] |

Ca—categorization: P—process; E—energy; C—combination of both.

In non-metallic mineral industries, rotary kilns are the most common type of kiln [21]. These industries are responsible for a substantial share of $CO_2$ emissions (the share is estimated at 4% of overall $CO_2$ production [17]).

Rotary kilns for clinker or lime production are able to combust solid, liquid, and even gaseous fuels or waste. In pre-calciners, where high temperatures are not required, fuels with lower heating values could be burnt [15,16]. Over the past decades, approximately 30% of the fuel combusted in non-metallic industries in EU28 has been solid. From 1990 to 2016, the shares of renewable fuels and waste fuels increased from 1 to 4.8% and from 0.2 to 7.7%, respectively [20]. These alternative fuels mainly replaced heavy fuel oil [21]. The alternative fuel used in cement plants is further presented in the section waste-to-energy.

Cement production can be divided into three processes [17]:

1. Raw material pre-processing;
2. Clinker production;
3. Cement production.

Ishak and Hashim [17] conclude several $CO_2$ mitigation methods:

- Technologies with high energy efficiency;

- Energy recovery;
- Alternative fuel combustion;
- Low-carbon cement production;
- $CO_2$ capture and storage.

According to this research, the optimization of lime and cement production is one of the most promising methods of $CO_2$ mitigation [17]. New, alternative technologies of clinker production are presented by Antunes et al. [23]. The authors deal with the option of the electrification of the whole production process. This technology could partially or even completely reduce $CO_2$ production within this process. However, clean technologies for electricity generation are needed.

RK optimization is presented by Engin and Ari [13]. The authors estimated that 40% of the total input energy is lost during the process. The kiln shell losses are about 15%, the cooler stack about 5%, and the highest amount of energy (approximately 20%) is lost through hot flue gases. For losses caused by hot flue gas and cooler stacks, the authors propose the installation of a waste heat recovery steam generator. It is estimated that about 4% of total input energy could be recovered. For losses caused by a kiln surface, a secondary shell is proposed. Approximately 12% of the total input energy could be saved from the kiln surface. From an economic point of view, the payback period of these two proposed solutions is about 1.5 years [13].

The energy supply requirements of cement and lime production are very high. Luo et al. [18] estimate the energy consumption of Portland cement production at about 110–120 kWh/ton. Approximately 10 to 15% of the energy is lost via the external surface of the rotary kiln. In order to minimize heat losses and optimize efficiency, a thermoelectric waste-heat-recovery system could be used. The principle of this system is in a thermoelectric generation unit implemented along with the kiln in a secondary shell. The influence of this system is estimated via a mathematical model. According to the results, approximately 33% of the lost energy could be recovered [18]. A complex overview of waste-heat recycling using thermoelectric generators is presented by Zaferani et al. [22]. In this paper, waste heat recovery from rotary cement kiln reactors is evaluated and discussed.

In order to increase lime production efficiency, Shahin et al. [19] propose several improvements for the entire process. According to the results from the mathematical model, a 10-min increase in the residence time in the preheater/kiln can decrease fuel consumption by approximately 2%. On the other hand, a five-minute increase in the residence time $\tau$ (hr) in the cooler shows similar results. The modeling of cement rotary kilns is reviewed by Saidur et al. [14], who state that a lot of RKs were built several decades ago. These kilns are not operated according to the best available techniques. Some kilns have the potential to double their operating capacity.

Each of the nine articles cited in this section can be categorized as process engineering. The research on RKs within clinker and lime production focuses on $CO_2$ emission mitigation, energy recovery or mitigation losses, and alternative fuels. The use of alternative fuel within building material production is further studied in Section 4.3. According to the mentioned articles, the research potential is substantial. The most promising is the use of alternative fuels instead of fossil fuels (it is estimated that alternative fuels can cover around 40% of the input energy). Older-type RKs, which are not operated according to new findings and standards, should make investments to minimize the negative impact of production. Due to high energy demand, the payback period of these investments is relatively low.

### 4.2. Solar Energy Utilization and Storage

RK operation (thermal treatment of particulate materials) is connected with a high consumption of fossil fuels. Thus, research activities of the second decade of the 21st century are focused on alternative sources of energy for RKs. The main recommendation is the use of solar energy. The cited articles and their motivation are presented in Table 2.

**Table 2.** Articles reviewed in Section 4.2.

| Nm. | Authors | Title | Year | Motivation—Remarks | Ca | Cit. |
|---|---|---|---|---|---|---|
| 1 | Meier A, Bonaldi E, Cella GM, Lipinski W, Wuillemin D, Palumbo R. | Design and experimental investigation of a horizontal rotary reactor for the solar thermal production of lime | 2004 | Environmental protection—Solar energy in lime and clinker production | C | [24] |
| 2 | Kaneko H, Miura T, Fuse A, Ishihara H, Taku S, Fukuzumi H, et al. | Rotary-Type Solar Reactor for Solar Hydrogen Production with Two-step Water Splitting Process | 2007 | Environmental protection—Hydrogen production in a solar rotary reactor | C | [25] |
| 3 | Mette B, Kerskes H, Drück H. | Process and Reactor Design for Thermo-Chemical Energy Stores | 2011 | Energy storage—Thermochemical heat storage | C | [26] |
| 4 | Neises M, Tescari S, de Oliveira L, Roeb M, Sattler C, Wong B. | Solar-heated rotary kiln for thermochemical energy storage | 2012 | Energy storage—Solar-heated RK for redox energy storage | C | [27] |
| 5 | Tescari S, Agrafiotis C, Breuer S, de Oliveira L, Puttkamer MN, Roeb M, et al. | Thermochemical Solar Energy Storage Via Redox Oxides: Materials and Reactor/Heat Exchanger Concepts | 2014 | Energy storage—Optimized RK for solar energy thermochemical storage | C | [28] |
| 6 | Grassmann H, Boaro M, Citossi M, Cobal M, Ersettis E, Kapllaj E, et al. | Solar Biomass Pyrolysis with the Linear Mirror II | 2015 | Environmental protection—Pyrolysis of agro-waste using solar energy | C | [29] |
| 7 | Solé A, Martorell I, Cabeza LF. | State of the art on gas–solid thermochemical energy storage systems and reactors for building applications | 2015 | Energy storage—Thermochemical heat storage | C | [30] |
| 8 | Alonso E, Pérez-Rábago C, Licurgo J, Fuentealba E, Estrada CA. | First experimental studies of solar redox reactions of copper oxides for thermochemical energy storage | 2015 | Energy storage—Solar energy utilization using thermal energy storage | C | [31] |
| 9 | Yadav D, Banerjee R. | A review of solar thermochemical processes | 2016 | Environmental protection—Solar thermochemical processes review | C | [32] |
| 10 | Zhang H, Baeyens J, Cáceres G, Degrève J, Lv Y. | Thermal energy storage: Recent developments and practical aspects | 2016 | Energy storage—Thermal energy storage | C | [33] |
| 11 | Alonso, E.; Gallo, A.; Roldán, M.I.; Pérez-Rábago, C.A. | Use of rotary kilns for solar thermal applications: Review of developed studies and analysis of their potential | 2017 | Environmental protection—RK and solar thermal applications | C | [3] |
| 12 | Koepf E, Alxneit I, Wieckert C, Meier A. | A review of high temperature solar driven reactor technology: 25 years of experience in research and development at the Paul Scherrer Institute | 2017 | Environmental protection—Solar reactors at Paul Scherrer Institute | C | [34] |
| 13 | Pelay U, Luo L, Fan Y, Stitou D, Rood M. | Thermal energy storage systems for concentrated solar power plants | 2017 | Energy storage—Solar energy utilization using thermal energy storage | C | [35] |
| 14 | Pan ZH, Zhao CY. | Gas–solid thermochemical heat storage reactors for high-temperature applications | 2017 | Energy storage—Thermochemical heat storage | C | [36] |
| 15 | Gallo A, Alonso E, Pérez-Rábago C, Fuentealba E, Roldán MI. | A lab-scale rotary kiln for thermal treatment of particulate materials under high concentrated solar radiation: Experimental assessment and transient numerical modeling | 2019 | Environmental protection—Particle heating using a high-flux solar simulator | C | [37] |
| 16 | Jiang K, Du X, Kong Y, Xu C, Ju X. | A comprehensive review on solid particle receivers of concentrated solar power | 2019 | Energy storage—Thermochemical heat storage using a centrifugal particle receiver | C | [38] |

Ca—categorization: P—process; E—energy; C—combination of both.

Solar energy utilization can be divided into photovoltaic (for electricity generation) and concentrated solar power (CSP). CSP systems are more advantageous in energy-intensive

and high-temperature processes. The typical CSP system consists of a field of heliostats that track the sun on two axes and concentrate solar radiation into the solar receiver. There are several reactor types—a rotary kiln reactor is one of them. The efficiency of solar energy utilization could be evaluated in different ways. Thus, the efficiency discussed in this review is provided with additional information on how it is calculated.

One of the most significant issues connected with solar energy is its unbalanced output in the daytime. In order to balance the output during the whole day, 24/7, the CSP system can be equipped with a thermal energy storage (TES) system [39].

TES supports the stability of energy delivery from solar systems; it can be implemented using three main approaches. The first one is called *sensible* and stores the heat by heating a heat carrier (without a phase change or chemical reaction). The second one is called *latent* and stores the heat using the phase change of the material. This approach uses the liquid–solid phase change and is more efficient than the sensible one. The third approach, called *thermochemical,* uses a chemical conversion. All three approaches could be used in a rotary kiln reactor.

Several thermochemical processes can be handled using CSP. Koepf et al. [34] summarize 25 years of research on high-temperature solar-driven reactor technology at the Paul Scherrer Institute in Switzerland. This review paper describes the research of five solar reactor projects (examples of rotating systems are mentioned in brackets) using CSP:

- The thermal reduction of zinc oxide (rotating solar reactor called ZIRRUS);
- The carbothermal reduction of zinc oxide (rotating vertical reactor);
- The partial reduction of ceria;
- The gasification of carbonaceous materials;
- The production of lime (rotary kiln).

Even though carbonaceous materials in this article are gasified in another type of reactor (two-cavity packed bed reactor), a rotary kiln reactor is also suitable for these processes.

Yadav et al. [32] review solar thermochemical processes. One of several outcomes is that solar methane reforming or lime production processes are more efficient in comparison with other thermochemical processes. Therefore, for commercialization, the authors recommend simpler processes with a single reaction such as the upgrading of carbon feed or the production of industrial commodities [32].

In the lab-scale research, a high-flux solar simulator (HFSS) is used to simulate solar radiation. HFSS consists of an adapted cinema projector. Gallo et al. [37] define kiln efficiency as the ratio of the energy absorbed by the particles to the overall thermal energy entering the facility. The authors show that the kiln efficiency is very low (ca. 1%) since most energy is used for kiln preheating. A significant amount of energy is reflected in the environment (ca. 35%). To avoid these losses, more batches are recommended. For continuous processes, higher efficiency is expected [37].

Due to the high energy demand for lime or clinker production, the use of solar energy in this industry is promising. However, so far, there have been only lab-scale technologies. Meier et al. [24] designed and tested a 10-kW solar horizontal rotary kiln reactor with a conical reaction chamber. The quality of the reaction (the calcination and the final product reactivity) was influenced by decomposition temperature, residence time, and feed rate. The process efficiency (the enthalpy of the calcination reaction divided by the solar energy input) was about 20%, and the production rate is approximately 1.3 kg/h. The reactor was operated in Switzerland on 24 sunny days for about 100 h. The quality of produced CaO meets industrial requirements [24].

Another technology is the combination of solar energy with the pyrolysis of waste materials. Grassmann et al. [29] presented a solar-heated rotary reactor. As a pyrolyzed material, agro-waste (wheat straw) was chosen. There exists a potential to increase the material energy density by about 50%. The lab-scale prototype capacity is 5 kg of agro-waste in approximately 2 h. Assuming that, during one sunny day, this prototype can transform about 20 kg of input material [29].

Hydrogen production is another application of a solar rotary reactor. Hydrogen is a perfect energy carrier with high potential not only in the automotive industry. However, its production demands a large amount of energy. Thus, clean ways of hydrogen production are auspicious. Kaneko et al. [25] showed the process enabled by reactive ceramics ($CeO_2$ and Ni, Mn ferrite). The rotary kiln reactor is situated in a quartz reaction cell. Half of this cell releases $O_2$, and half generates $H_2$. The outer surface of the rotary kiln is made of reactive ceramics. The laboratory-scale reactor shows that the production of hydrogen within a solar-heated rotary reactor is possible. The production rate of $H_2$ is 0.02 m$^3$/min during the process [25].

Materials that enable a thermochemical reaction suitable for the energy storage reaction are called thermochemical materials (TCM). So far, thermochemical heat storage has been tested only at the laboratory or pilot scale [33]. However, due to its reliability, low cost, and easy implementation, operating plants mostly use the sensible heat storage approach. It also enables good feedback for experimental research [35].

The advantages of the thermochemical heat storage approach are its high energy density, stability in the long term, and ambient storage temperature [36]. Reactors for thermochemical heat storage, such as rotary solar kilns, can be divided into open and closed or separate and integrated reactors [30]. The separated reactors transport thermochemical material (TCM) after the reaction to the storage vessel, while in the integrated reactors, the reaction occurs directly in the storage vessel. Thus, the integrated approach has a lower pump power consumption. On the other hand, the separated approach using more storage vessels can split the whole process into smaller parts. As a result, the reactor can be smaller with lower pressure drops and easier control [26].

Thermo-chemical storage systems can utilize not only solar energy. They can be adapted to cogeneration systems, as well [26].

The thermo-chemical energy storage approach was tested using a rotary solar reactor kiln by Alonso et al. [31]. As a reaction, a $CuO/Cu_2O$ redox cycle was chosen. About an hour-long cycle uses an air and argon atmosphere. The experiment was carried out in a rotary kiln with the direct absorption of reactive particles. The authors believe that this configuration ensures higher operation temperatures compared to other types of reactors. It was proved that using copper oxide for thermochemical energy storage is suitable [31].

In 2012, Neises et al. [27] proved that the solar-heated rotary kiln is feasible for redox energy storage. Compared to the results from the packed bed, the reaction time using a rotary kiln for the entire cycle of the high reversibility of $Co_3O_4 + Al_2O_3$ is significantly lower (about 50 min). The authors state that the rotary kiln is able to store about 400 kJ/kg in the energy carrier. On the other hand, due to insufficient mixing in the rotary kiln, only half of the material reacts [27].

The use of rotary kilns is advantageous for thermo-chemical heat storage due to the ability to obtain homogeneous temperature, good heat and mass transfer, and a wide range of applications. On the other hand, the rotary kilns show high energy losses (a significant amount of energy is wasted on kiln preheating). To increase the efficiency, Tescari et al. [28] developed an optimized design of the rotary kiln where the particles enter the chamber from the opposite side of the solar concentrator. Further, this rotary kiln was enhanced with four straight flights to improve mixing inside the chamber [28].

Another implementation of the rotary kiln in thermochemical heat storage systems is the centrifugal particle receiver. The particles are forced against the wall due to centrifugal forces. The Froude number is equal to 1. They create a thin but optically dense layer on the whole inner surface of the receiver. The particles ensure heat transfer and heat storage at the same time. Axial movement along the kiln is ensured by gravitation [38].

In agreement with Alonso et al. [3], the main limitations and recommendations for rotary reactor use within solar systems can be concluded. To meet the requirements for high-temperature applications, expensive and not always available materials are needed. Further investigation should be focused on industrial applications such as the metallurgy of certain materials (lead, copper, or iron processing).

Approximately half of the 16 articles cited in this section deal with heat energy storage. Thermo-chemical storage systems are a promising path for future development, especially if their advantages are connected with cogeneration systems. However, a current level of knowledge enables only a low amount of stored energy. If we compare the heat stored in some standard fuel (let us assume the average LHV of lignite is 15 MJ/kg) and in some carriers feasible in solar energy storage systems (Neises et al. [27] reached 400 kJ/kg), the differences are still substantial. Thus, current solar energy storage systems operate on a laboratory or pilot scale. Mostly, there is a latent type of energy storage system. The advantages of these systems are their reliability, low cost, and easy implementation. It is expected that these systems will be able to cover at least a part of the fossil fuel-based systems in the future.

The rest of cited articles deal with solar energy utilization within RK operations, such as pyrolysis or lime and clinker production. The main motivation of this research is to utilize and store clean and cheap solar energy, replace standard fossil fuels, and minimize the negative environmental impact of RK operation.

### 4.3. Waste-to-Energy including Bio-Based Material

The waste-to-energy approach is one of the most common research topics. Using the keyword in the ScienceDirect database shows over half a million articles. In this review, 86/131 articles were related to this topic. Table 3 presents articles that are not categorized in Sections 4.3.1–4.3.3. The waste-to-energy approach aims to utilize the energy content of waste that cannot be recycled by electricity generation and heat utilization. This approach is widespread, so several sub-topics (such as pyrolysis—24/131 articles; sewage sludge treatment—21 articles; and waste tires treatment—18 articles) are discussed in separate sections.

**Table 3.** Articles reviewed in Section 4.3.

| Nm. | Authors | Title | Year | Motivation—Remarks | Ca. | Cit. |
|---|---|---|---|---|---|---|
| 1 | Jenkins BM, Baxter LL, Miles TR, Miles TR. | Combustion properties of biomass | 1998 | New fuels—Biomass co-incineration | C | [40] |
| 2 | Sami M, Annamalai K, Wooldridge M. | Co-firing of coal and biomass fuel blends | 2001 | New fuels—Biomass co-incineration | C | [41] |
| 3 | McKendry P. | Energy production from biomass (part 2): conversion technologies | 2002 | New fuels—Biomass co-incineration | C | [42] |
| 4 | Conesa JA, Fullana A, Font R. | Thermal decomposition of meat and bone meal | 2003 | New fuels—Meat and bone meal disposal within cement RK | P | [43] |
| 5 | Lemieux P, Stewart E, Realff M, Mulholland JA. | Emissions study of co-firing waste carpet in a rotary kiln | 2004 | New fuels—End-of-life carpet treatment within cement RK | C | [44] |
| 6 | Hoffmann G, Schirmer M, Bilitewski B, Kaszás/Savos M. | Thermal treatment of hazardous waste for heavy metal recovery | 2007 | Optimization—Heavy metal recovery using RK | C | [45] |
| 7 | Vermeulen I, Van Caneghem J, Block C, Baeyens J, Vandecasteele C. | Automotive shredder residue (ASR): Reviewing its production from end-of-life vehicles (ELVs) and its recycling, energy, or chemicals' valorization | 2011 | New fuels—Automotive shredder residue treatment within RK | C | [46] |
| 8 | Ariyaratne WKH, Melaaen MC, Eine K, Tokheim LA. | Meat and bone meal as a renewable energy source in cement kilns: investigation of optimum feeding rate | 2011 | New fuels—Meat and bone meal disposal within cement RK | P | [47] |
| 9 | Nielsen, A. R., Larsen, M. B., Dam-Johansen, K., Glarborg, P. | Combustion of large solid fuels in cement rotary kilns | 2012 | New fuels—Alternative fuels in cement production | C | [2] |
| 10 | Cascarosa E, Gea G, Arauzo J. | Thermochemical processing of meat and bone meal: A review | 2012 | New fuels—Meat and bone meal disposal within cement RK | P | [48] |

**Table 3.** *Cont.*

| Nm. | Authors | Title | Year | Motivation—Remarks | Ca. | Cit. |
|---|---|---|---|---|---|---|
| 11 | Fernando R. | Cofiring high ratios of biomass with coal | 2012 | New fuels—Biomass co-incineration | C | [49] |
| 12 | Rahman A, Rasul MG, Khan MMK, Sharma S. | Impact of Alternative Fuels on the Cement Manufacturing Plant Performance: An Overview | 2013 | New fuels—Alternative fuels in cement production | P | [16] |
| 13 | Ishak SA, Hashim H. | Low carbon measures for cement plant—a review | 2015 | Environmental protection—$CO_2$ mitigation in cement production | P | [17] |
| 14 | Lombardi L, Carnevale E, Corti A. | A review of technologies and performances of thermal treatment systems for energy recovery from waste | 2015 | New fuels—Approaches to thermal waste treatment for energy recovery | E | [50] |
| 15 | Holcim (Deutschland) GmbH | Umweltbericht | 2015 | Optimization—Environmental measures of the Dotternhausen cement plant | P | [51] |
| 16 | Jaap Koppejan; Kay Schaubach; Janet Witt; Daniela Thrän. | Production of Solid Sustainable Energy Carriers from Biomass by Means of Torrefaction | 2015 | New fuels—Biomass co-incineration | C | [52] |
| 17 | Horsley C, Emmert MH, Sakulich A. | Influence of alternative fuels on trace element content of ordinary Portland cement | 2016 | New fuels—Waste-derived fuels in cement production | P | [53] |
| 18 | Bani-Hani EH, Hammad M, Matar A, Sedaghat A, Khanafer K. | Numerical analysis of the incineration of polychlorinated biphenyl wastes in rotary kilns | 2016 | New fuels—Hazardous waste treatment in RKs | E | [54] |
| 19 | Huber F, Blasenbauer D, Mallow O, Lederer J, Winter F, Fellner J. | Thermal co-treatment of combustible hazardous waste and waste incineration fly ash in a rotary kiln | 2016 | Optimization—Addition of flying ash into regular solid fuel as a way of its disposal | P | [55] |
| 20 | Hong J, Chen Y, Wang M, Ye L, Qi C, Yuan H, et al. | Intensification of municipal solid waste disposal in China | 2017 | New fuels—Municipal solid waste treatment optimization | E | [56] |
| 21 | Makarichi L, Jutidamrongphan W, Techato K. | The evolution of waste-to-energy incineration: A review | 2018 | New fuels—Municipal solid waste treatment | E | [57] |
| 22 | Liu J, Zhang H, Yao Z, Li X, Tang J. | Thermal desorption of PCBs contaminated soil with calcium hydroxide in a rotary kiln | 2019 | New fuels—Polychlorinated-biphenyl-contaminated soil cleaning using RK | P | [58] |
| 23 | Wang J, Shen J, Ye D, Yan X, Zhang Y, Yang W, et al. | Disinfection technology of hospital wastes and wastewater: Suggestions for disinfection strategy during coronavirus Disease 2019 (COVID-19) pandemic in China | 2020 | New fuels—Hospital and hazardous waste treatment using RK | E | [59] |

Ca—Categorization: P—process; E—energy; C—combination of both.

The advantage of rotary kilns used within waste-to-energy systems is their ability to process any waste, including liquid waste. Compared with mobile grates, the rotary kiln withstands higher temperatures (approximately 1400 vs. 1250 °C) [50]. Mobile grates have a higher capacity, easier mixing with an oxidizer, smaller space requirements, and easier heat utilization. The share of the grate or fluidized bed technologies in municipal solid waste incineration plants is dominant (in China, more than 97% [56]). In the EU, in 2012, 88% of Municipal solid waste (MSW) was disposed of with moving grate technology (in Germany, it was 94%, and in the US, 76%) [57]. The rotary kilns are more suitable for dangerous waste treatment such as oil sludge, oily textiles, waste oil, hospital waste, or end-of-life tire treatment. End-of-life tire treatment is discussed in detail in Section 4.3.3.

The combustion of renewable fuels (e.g., biomass) that replace fossil fuels decreases greenhouse gas emissions. On the other hand, waste-derived fuels, such as tires, plastics, or industrial waste, that do not reduce emission intensity, could not be considered renewable fuels [51]. However, their disposal via combustion or other thermal processes is still

a good solution for volume reduction, hazardous and harmful material disposal, and energy utilization.

The potential of the greenhouse effect of neutral fuels replacement in clinker production is estimated to be about 40% (40% of non-renewable fuels could be replaced), which means an approximate $CO_2$ production of 0.53 tons per ton of clinker (non-renewable carbon-based fuels such as coal or petroleum coke produce about 0.85 ton $CO_2$ per ton of clinker) [17]. However, the increased amount of biomass in fuel can influence the quality of the clinker since the ash residuals remain in this final product. Thus, the mineral composition of the alternative fuel has to be controlled [51].

A very complex work focused on the replacement of fossil fuels with alternative fuels in cement rotary kilns is a Ph.D. thesis by Nielsen [2]. The studied alternative fuels are refuse-derived fuel, tire-derived fuel, meat and bone meal, waste wood, sewage sludge, paper, and plastics. Mixing and combustion experiments were carried out in a pilot-scale rotary kiln. One of the author's conclusions is that the process stability and the product quality are influenced by the different physical and chemical properties of alternative fuels (in comparison with traditional fossil fuels). Thus, new equipment must be implemented within the rotary kiln, or current kiln systems must be modified.

Moreover, the author focused on the utilization of large particles with no need for pre-shredding. It was found that the devotilization of large wood or tire particles with a thickness of about 20 mm is relatively fast (about 2 min). On the other hand, the oxidization of char is a prolonged process, which may affect the final product quality. If minor compounds of unburned carbon get into the clinker, its quality may decrease. To avoid this, the author created models of char devolatilization. Char oxidation is influenced by rotational speed or process temperature and particle size or shape, filling ratio, bed oxygen concentration, etc. [2].

Horsley et al. [53] state that waste-derived fuel (WDF) processing is more economically favorable for already-existing plants than building completely new ones. Due to the high energy consumption of cement kilns, it is advantageous to treat WDF within these processes. Due to high temperatures and long residence time, a wide range of waste and hazardous materials could be combusted in cement kilns. Rahman et al. [16] summarize the literature to describe the influence of alternative fuels on the process parameters. Many alternative fuels are combusted within cement or lime production, such as waste oil, tires, plastics, industrial and household waste, agricultural waste, wood chips, other biomass, sewage sludge, etc. Results reveal that alternative fuels could reduce $NO_x$, $SO_2$ emissions, or dust [16].

Another source of WDF is end-of-life vehicles. The waste of 25% of these vehicles cannot be recycled. Vermeulen et al. [46] study the automotive shredder residue treatment, including incineration within a rotary kiln or co-incineration within a cement rotary kiln. The authors believe that its co-incineration with MSW or sewage sludge is suitable and already used. However, its disposal within a cement rotary kiln is suitable only at a low percentage. A higher amount requires a previous upgrade due to ash formation, clogging, or mercury content in the clinker. Another research paper studies the co-incineration of waste carpets within a cement rotary kiln. The results reveal that about 30% of total energy input could be covered with waste carpet incineration. During the experiment, a negligible difference in CO emission production was found. On the other hand, NO emissions increased due to the nitrogen composition of nylon fibers [44].

Hazardous waste is waste that could be incinerated in a rotary kiln. Polychlorinated biphenyls are used in electrical appliances, and their disposal is quite problematic due to their resistance to high temperatures. Rotary kilns could be used for the thermal desorption of polychlorinated-biphenyl-contaminated soil using calcium hydroxide. An experiment with blank soil and 1% $Ca(OH)_2$ added to a rotary kiln was carried out. According to the results, calcium hydroxide strengthens the removal, dechlorination, and detoxication of polychlorinated biphenyls [58].

A numerical analysis of polychlorinated-biphenyl waste treatment within an RK using the MATLAB routine is presented by Bani-Hani et al. [54]. A dynamic model simulates the mass, energy, and concentration of species during combustion. The model can simulate different process conditions in the rotary kiln for different waste compositions. According to the simulation results, the authors recommend optimal operating conditions for polychlorinated-biphenyl waste incineration [54].

The experiment with the disposal of municipal solid waste incineration (MSWI) fly ash in a rotary kiln was investigated. It was found that the addition of MSWI fly ash into the regular fuel (in this case, hazardous waste) does not affect the rotary kiln operation. More than 90% of MSWI fly ash was transferred to the bottom ash during the incineration process. The legal limits for non-hazardous waste landfills were complied with [55].

Heavy metal recovery from hazardous waste in the rotary kiln was investigated. The principle of this method is a thermal treatment in a sub-stochiometric atmosphere in the rotary kiln. Liquid and paste materials could be treated this way. The final product contains high concentrations of heavy metal that could be further recycled [45].

Due to the current pandemic caused globally by COVID-19, there is a strong effort towards disinfection technology and the treatment of hospital and hazardous waste. Wang et al. [59] reviewed several methods of hazardous waste disposal. Rotary kilns are suitable if the amount of generated waste is enormous. However, the high dust composition in the flue gas and high investment and maintenance costs have to be considered [59].

Animal byproducts must also be considered hazardous; therefore, they have to be combusted in order to prevent infection. Abattoir waste can be divided into meat scraps and bone, which are usually processed to produce tallow and meat and bone meal (called MBM) [48]. Part of MBM can be used for animal feed, and the rest is combusted (energy content is in the range of 13 to 30 MJ/kg) or landfilled [43]. However, the high composition of chloride limits the maximum amount of MBM combusted during cement production. MBM can also contaminate the clinker with phosphates. Thus, the clinker quality has to be controlled [47].

Problems connected with municipal solid waste incineration within regular incineration power plants could be solved by MSW pre-treatment using pyrolysis in the rotary kiln. This approach is further discussed in Section 4.3.1. Also promising is the mixing of alternative fuels to strengthen their advantages and weaken their disadvantages. Several papers are focused on alternative fuel mixing. However, no paper has been found with a focus on the influence of alternative fuel composition on particulate matter formation. Finding a good mix of alternative fuels (such as sewage sludge, municipal solid waste, biomass, etc.) can optimize fouling and slagging within rotary kiln processes. Moreover, the more facilities suitable to utilize any type of waste (pre-treated or mixed in the proper order), the easier the distribution, the lower the transport cost, and the lower the overall $CO_2$ production. Thus, this task is crucial considering the WtE approach.

Biomass is not considered waste. However, there are several similarities to waste regarding its energy utilization in the rotary kiln. Thus, biomass energy and material utilization are included in the WtE section. Biomass combustion causes several problems. Ash composition varies during biomass incineration depending on the biomass type [49]. A higher composition of alkali metal and chlorine can increase fouling and slagging, which are responsible for lower heat transfer, lower overall efficiency, higher corrosion, and more frequent maintenance, resulting in higher costs [40,52]. Compared to incineration using only coal, co-firing 22% of biomass increases the rate of corrosion two-fold [41].

Untreated biomass usually contains a high amount of moisture, between 25% and 50%. Therefore, the energy content is low, and storage requirements are high. The high moisture also complicates the storage since the material degrades and releases gas and heat [49]. Low energy content or storage difficulties could be solved by several technologies of pre-treatment, such as torrefaction, drying, pelletizing [52], gasification, or pyrolysis [42].

Waste-to-energy is a complex approach that connects energy and process engineering within RK applications. The proof is that the categorization into energy (5/23) or process

(9/23) engineering or their combination (9/23) is relatively equal. The main research areas in WtE research connected with RKs represent a search for different types of waste suitable for combustion within RKs, the optimization of RK operation, and negative environmental impact mitigation. Various wastes, including hazardous ones, can be combusted within an RK. The main motivation is to find an optimal waste mixture and minimize its negative impact.

### 4.3.1. Pyrolysis and Gasification

Pyrolysis and gasification-based research connected with RK is substantial; therefore, this subsection of the present paper is devoted to these technologies. Just 24 of 131 articles connected with this topic are presented in Table 4. Any carbon-based material could be treated using a thermochemical process called pyrolysis. The principle is to heat the material in an inert atmosphere (absence of oxygen). The material is subjected to an endothermic chemical reaction while its molecules are separated. The final products are pyrolytic gas, oil, and biochar. The share of the yield of these products is related to the process conditions such as temperature and residence time.

**Table 4.** Articles reviewed in Section 4.3.1.

| Nm. | Authors | Title | Year | Motivation—Remarks | Ca. | Cit. |
|---|---|---|---|---|---|---|
| 1 | Klose E, Born M. | Partial gasification of lignite coke with steam in a rotary kiln for activated carbon production | 1985 | New fuels—MSW pyrolysis within an RK | C | [60] |
| 2 | Li AM, Li XD, Li SQ, Ren Y, Shang N, Chi Y, et al. | Experimental studies on municipal solid waste pyrolysis in a laboratory-scale rotary kiln | 1999 | New fuels—Waste and biomass pyrolysis | C | [61] |
| 3 | Malkow T. | Novel and innovative pyrolysis and gasification technologies for energy efficient and environmentally sound MSW disposal | 2004 | Optimization—Modeling of RK processes | C | [62] |
| 4 | Descoins N, Dirion J-L, Howes T. | Solid transport in a pyrolysis pilot-scale rotary kiln: preliminary results—stationary and dynamic results | 2005 | Optimization—Modeling of RK processes | P | [63] |
| 5 | Van Puyvelde DR. | Simulating the mixing and segregation of solids in the transverse section of a rotating kiln | 2006 | New fuels—Biomass upgrading using torrefaction | P | [64] |
| 6 | Kiel JHA, Verhoeff F, Gerhause H, Meulemann B. | $BO_2$-technology for biomass upgrading into solid fuel | 2008 | New fuels—Ways of microwave pyrolysis, including RKs | C | [65] |
| 7 | Yuen FK, Hameed BH. | Recent developments in the preparation and regeneration of activated carbons by microwaves | 2009 | New fuels—Ways of biomass torrefaction, including a rotary drum facility | C | [66] |
| 8 | Acharya B, Sule I, Dutta A. | A review on advances of torrefaction technologies for biomass processing | 2012 | New fuels—Steam gasification within an RK | C | [67] |
| 9 | Iovane P, Donatelli A, Molino A. | Influence of feeding ratio on steam gasification of palm shells in a rotary kiln pilot plant. Experimental and numerical investigations. | 2013 | New fuels—Ways of biomass torrefaction including a rotary drum facility | C | [68] |
| 10 | Batidzirai B, Mignot APR, Schakel WB, Junginger HM, Faaij APC. | Biomass torrefaction technology: Techno-economic status and future prospects | 2013 | New fuels—Waste, paper, plastics, and biomass pyrolysis | C | [69] |

**Table 4.** *Cont.*

| Nm. | Authors | Title | Year | Motivation—Remarks | Ca. | Cit. |
|---|---|---|---|---|---|---|
| 11 | Chen D, Yin L, Wang H, He P. | Pyrolysis technologies for municipal solid waste: A review | 2014 | New fuels—Ways of biomass torrefaction, including a rotary drum facility | C | [70] |
| 12 | Nhuchhen D, Basu P, Acharya B. | A Comprehensive Review on Biomass Torrefaction | 2014 | New fuels—Pyrolysis of biogas digestate within an RK | C | [71] |
| 13 | Monlau F, Sambusiti C, Antoniou N, Barakat A, Zabaniotou A. | A new concept for enhancing energy recovery from agricultural residues by coupling anaerobic digestion and pyrolysis process | 2015 | New fuels—Ways of biomass torrefaction, including a rotary drum facility | C | [72] |
| 14 | Eseyin AE, Steele PH, Pittman Jr. CU. | Current Trends in the Production and Applications of Torrefied Wood/Biomass—A Review | 2015 | New fuels—MSW pyrolysis | C | [73] |
| 15 | Czajczyńska D, Anguilano L, Ghazal H, Krzyżyńska R, Reynolds AJ, Spencer N, et al. | Potential of pyrolysis processes in the waste management sector | 2017 | New fuels—Pyrolysis of biomass in an RK | C | [74] |
| 16 | Promdee K, Chanvidhwatanakit J, Satitkune S, Boonmee C, Kawichai T, Jarernprasert S, et al. | Characterization of carbon materials and differences from activated carbon particle (ACP) and coal briquettes product (CBP) derived from coconut shell via rotary kiln | 2017 | New fuels—Biomass pyrolysis within an indirect-heated RK | C | [75] |
| 17 | Babler MU, Phounglamcheik A, Amovic M, Ljunggren R, Engvall K. | Modeling and pilot plant runs of slow biomass pyrolysis in a rotary kiln | 2017 | New fuels—Ways of microwave pyrolysis, including RK | C | [76] |
| 18 | Beneroso D, Monti T, Kostas ET, Robinson J. | Microwave pyrolysis of biomass for bio-oil production: Scalable processing concepts | 2017 | New fuels—Ways of biomass torrefaction, including rotary drum facility | C | [77] |
| 19 | Ribeiro J, Godina R, Matias J, Nunes L. | Future Perspectives of Biomass Torrefaction: Review of the Current State-Of-The-Art and Research Development | 2018 | New fuels—Waste and biomass pyrolysis | C | [78] |
| 20 | Campuzano F, Brown RC, Martínez JD. | Auger reactors for pyrolysis of biomass and wastes | 2019 | New fuels—Pre-treatment of MSW within an RK | C | [79] |
| 21 | Kuo W-C, Lasek J, Słowik K, Głód K, Jagustyn B, Li Y-H, et al. | Low-temperature pre-treatment of municipal solid waste for efficient application in combustion systems | 2019 | New fuels—Production of charcoal in different types of kilns | C | [80] |
| 22 | Rodrigues T, Braghini Junior A. | Charcoal: A discussion on carbonization kilns | 2019 | New fuels—Biomass gasification within an RK | C | [81] |
| 23 | Ren J, Cao J-P, Zhao X-Y, Yang F-L, Wei X-Y. | Recent advances in syngas production from biomass catalytic gasification: A critical review on reactors, catalysts, catalytic mechanisms and mathematical models | 2019 | New fuels—Biomass gasification within an RK | C | [82] |
| 24 | Ren J, Liu Y-L, Zhao X-Y, Cao J-P. | Methanation of syngas from biomass gasification: An overview | 2020 | New fuels—MSW pyrolysis within an RK | C | [83] |

Ca—Categorization: P—process; E—energy; C—combination of both.

Gasification is a very similar process where a non-stoichiometric amount of oxygen occurs in order to combust a part of the input material, which is the source of heat for gasification. However, compared to pyrolysis, gasification requires higher temperatures (over 800 °C), and the main product is gas (mainly used for electricity generation).

A rotary kiln is not the only facility for pyrolysis and gasification. Table 5 shows the main types of pyrolysis, including the typical residence time, temperature ranges, suitable reactors, and final products.

**Table 5.** Types of pyrolysis, typical conditions, and products. Based on [79].

| Type of Pyrolysis | Residence Time | Temp. (°C) | Technology (Commonly Used Reactors) | Main Products |
|---|---|---|---|---|
| Fast | Short | 400–600 | Fluidized bed, spouted bed, auger | Liquid |
| Intermediate | From seconds to min | 400–600 | Auger, rotary kiln, fixed bed | Liquid, solid, and gas |
| Slow—torrefaction | From min to h | 250–350 | Auger, rotary kiln, fixed bed | Solid |
| Slow—carbonization | From h to days | 300–500 | Auger, rotary kiln, fixed bed | Solid |

Malkow [62] states that, compared to the regular combustion of MSW, producing hazardous emissions and harmful process residues, pyrolysis is a more suitable technology for MSW disposal. The influence of pyrolysis conditions during waste processing, such as temperature, residence time, etc., on the chemical and mineralogical composition of final products has been investigated by several research groups [70,74]. According to the experiments carried out by Li et al. [61] using municipal solid waste (paper, plastics, wood, etc.) pyrolysis within a rotary kiln, there are several relationships between different types of pyrolysis:

- The fast type of pyrolysis requires less reaction time and produces more gas in comparison with the slow type;
- HHV and the composition of produced gas vary during pyrolysis;
- The higher the temperature, the higher the production of small molecules, and the lower the production of large molecules;
- HHV, at first, increases with temperature, and then decreases;
- The moisture of the input material has a stronger influence on the product quality than the material size [61].

A simulation of the mixing and segregation of solids in a rotary kiln was investigated by Puyvelde [64] and Descoins [63]. In addition, a pilot-scale experiment of coconut shell conversion in a rotary kiln is carried out by Promdee et al. [75]. The final products were activated carbon particles (ACP) and coal briquette products (CBPs). The main findings were as follows:

- The quality of the final product meets chemical and physical requirements;
- The higher the temperature during the process, the better the quality of the end product;
- For comparison with other technologies, the HHV of the product is in the range of 22 to 26.5 MJ/kg [75].

A higher percentage of non-treated MSW incineration within regular power plants is undesirable due to increased slagging or fouling. This could be solved by MSW pre-treatment using so-called torrefaction (a type of slow pyrolysis using low temperatures) in a rotary kiln [80].

Rodrigues and Junior [81] identified 21 types of kilns (including RKs) where charcoal is produced from various types of waste (wood, coconut shells, straw, fermentation residues, etc.). The authors state that most charcoal is still produced in low-technology kilns with low quality. Several aspects of the improvement of charcoal production were presented [81].

Europe, nowadays, suffers from a severe bark beetle calamity. Due to this, there is a vast amount of wood biomass that has to be treated. The bulk density of torrefied wood pellets can reach one-third of coal [65].

RKs could be used for the steam gasification of different types of material. Ren et al. [82,83] presented the main advantages and disadvantages of rotary kilns for biomass gasification. They can process input materials with different properties, flexible loading, large-scale options, simple construction, high reliability, and relatively low investment costs.

On the other hand, rotary kilns present difficulty in controlling the initial temperature, low heat exchange capacity, low heat efficiency, and high maintenance costs [83].

The steam gasification of palm shells in the rotary kiln was investigated using a numerical model and experiment. It was revealed that the steam-to-biomass ratio has a significant influence on the process quality [68]. Other steam gasification experiments using the rotary kiln were carried out with lignite briquettes [60].

Babler et al. [76] presented a model and pilot-scale experiments processing wood chips in an indirectly heated rotary kiln. The final products were pyrolysis gas and biocoal. The model was in good agreement with the obtained experimental data. In addition, the optimal conditions for maximizing the pyrolysis yield were found.

There is high research potential in combining two or more different technologies into a new one. A good example could be the pyrolysis of digestate from biogas power plants which generally transform biomass waste into digestate and biogas using anaerobic digestion. Biogas is combusted in order to produce electricity and heat. Part of the electricity is used within the process, while the rest is sold to the public grid. The heat that is not used within the process and cannot be sold (usually due to low local demands or during the summer period) is usually wasted. Monlau et al. [72] investigated the feasibility of a combination of anaerobic digestion and pyrolysis. The testing facility was a quartz rotary kiln reactor.

The results revealed that waste heat produced within regular biogas plant processes could cover the need for digestate drying. The LHV of gained syngas was 15.7 MJ/Nm$^3$, while the HHV of gained pyrolysis oil after water extraction was 23.5 MJ/kg. This combined process has the potential to increase the electricity production of biogas power plants [72]. Moreover, due to new additional products that could be sold in the market, the economy of biogas power plants could be improved. However, first, a detailed investigation of the impact of this solution should be made. The research should focus on the impact on transport costs, electricity sale revenue, investment costs, payback period, etc.

Another type of pyrolysis that could be carried out in a rotary kiln is pyrolysis of a slow type; so-called torrefaction. The typical conditions are an inert environment, a temperature range of 200–300 °C, atmospheric pressure during this process, hydrophobicity, grindability, and an increase in the energy density of the feedstock. A review of biomass torrefaction in the rotary kiln is presented by Nhuchhen et al. [71], Acharya et al. [67], and others [69,73,78].

Compared to the conventional pyrolysis processes, faster and higher product quality could be reached using microwave pyrolysis. In conventional facilities, the heat is located outside of the material (the heat is distributed via conduction and/or convection), while in microwave facilities, the energy is supplied by microwaves directly to the material. Yuen et al. [66] report that the conversion (from electrical to thermal energy) efficiency is approximately 50% for 2450 MHz and 85% for 915 MHz.

Beneroso et al. [77] believe that the advantage of the use of a rotary kiln for microwave pyrolysis is the lack of a need for material compression. Thus, there is a better mass transfer of volatiles from the material to the environment. On the other hand, this technology requires a long residence time and suffers from uneven and low energy distribution.

Rotary kilns are suitable for pyrolysis and gasification processes. Just 24 of 131 articles are related to the process of material enhancement. Thus, most of them were categorized as a combination of the process and energy industries. The output product, such as gas, oil, or biochar, could be used as fuel to replace standard fossil fuels. Waste distribution into WtE plants or other suitable facilities should also be taken into consideration. Thus, the more plants that are able to incinerate the waste, the easier the distribution, with lower transport costs and lower emission production. However, the combustion of non-treated or non-separated MSW within regular incineration plants is questionable because these plants were built to combust coal. MSW combustion in these plants is problematic due to unstable content responsible for increased fouling, slagging, and corrosion. Thus, there appears to be a strong potential for the thermal pre-treatment of MSW (such as pyrolysis) in order to be combusted in regular power plants. Another promising technology is a combination of

biogas production in the biogas power plant and digestate pyrolysis. This approach has a solid potential to increase biogas power plant efficiency.

Compared to the fluidized bed, the advantages of RK are flexibility, capacity, reliability, and relatively low investment costs. On the other hand, their disadvantages are being difficult to control and having low heat efficiency and rather high maintenance costs. The potential research area is the search for a material eligible for pyrolysis in order to replace standard fuels.

### 4.3.2. Sewage Sludge Treatment

Municipal and industrial wastewater is treated in wastewater treatment plants (WTPs). Sewage sludge is a mud-like byproduct of this process. This waste in the EU is estimated at 11.25 million tons, and half is used in agricultural applications [43]. Developed countries produce more sewage sludge than developing ones. Thus, it is expected that the amount of sewage sludge will increase [84]. The articles concerned (21/131) are presented in Table 6.

**Table 6.** Articles reviewed in Section 4.3.2.

| Nm. | Authors | Title | Year | Motivation—Remarks | Ca. | Cit. |
|-----|---------|-------|------|--------------------|-----|------|
| 1 | Werther J, Ogada T. | Sewage sludge combustion | 1999 | New fuels—Sewage sludge incineration technologies, including RKs | C | [85] |
| 2 | Mininni G, Braguglia CM, Marani D. | Partitioning of Cr, Cu, Pb and Zn in sewage sludge incineration by rotary kiln and fluidized bed furnaces | 2000 | New fuels—Comparison of fluidized beds and rotary kilns within wastewater incineration processes | C | [86] |
| 3 | Shen L, Zhang D-K. | An experimental study of oil recovery from sewage sludge by low-temperature pyrolysis in a fluidised-bed | 2003 | New fuels—Sewage sludge and MSW mixture pyrolysis | C | [87] |
| 4 | Wolski N, Maier J, Hein KRG. | Fine particle formation from co-combustion of sewage sludge and bituminous coal | 2004 | New fuels—Sewage sludge and lignite mixture combustion | C | [88] |
| 5 | Shen L, Zhang D. | Low-temperature pyrolysis of sewage sludge and putrescible garbage for fuel oil production | 2005 | New fuels—Sewage sludge and putrescible garbage mixture pyrolysis | C | [89] |
| 6 | Podedworna J, Umiejewska K. | The technology of sludge | 2008 | New fuels—Sewage sludge treatment within cement production | P | [90] |
| 7 | Pettersson A, Åmand L-E, Steenari B-M. | Leaching of ashes from co-combustion of sewage sludge and wood—Part I: Recovery of phosphorus | 2008 | New fuels—Sewage sludge and wood mixture combustion | C | [91] |
| 8 | LeBlanc RJ et al. | Global atlas of excreta, wastewater sludge, and biosolids management: moving forward the sustainable and welcome uses of a global resource | 2009 | New fuels—Sewage sludge treatment—rotary dryer and rotary kiln suitability | C | [84] |
| 9 | Cohen Y. | Phosphorus dissolution from ash of incinerated sewage sludge and animal carcasses using sulphuric acid | 2009 | Optimization—Phosphate recovery within WWT processes using an RK | P | [92] |
| 10 | Werle S, Wilk RK. | A review of methods for the thermal utilization of sewage sludge: The Polish perspective | 2010 | New fuels—Sewage sludge treatment within cement production | P | [93] |

**Table 6.** *Cont.*

| Nm. | Authors | Title | Year | Motivation—Remarks | Ca. | Cit. |
|---|---|---|---|---|---|---|
| 11 | Bennion EP, Ginosar DM, Moses J, Agblevor F, Quinn JC. | Lifecycle assessment of microalgae to biofuel: Comparison of thermochemical processing pathways | 2015 | New fuels—Usage of RKs within WWT technology | C | [94] |
| 12 | Horsley C, Emmert MH, Sakulich A. | Influence of alternative fuels on trace element content of ordinary Portland cement | 2016 | New fuels—Waste-derived fuels in cement production | P | [53] |
| 13 | Naamane S, Rais Z, Taleb M. | The effectiveness of the incineration of sewage sludge on the evolution of physicochemical and mechanical properties of Portland cement | 2016 | New fuels—Sewage sludge treatment within cement production | P | [95] |
| 14 | ASHDEC | Industrial Process and Pilot Plant | 2016 | Optimization—Phosphate recovery within WWT processes using an RK | P | [96] |
| 15 | Ahmad T, Ahmad K, Alam M. | Sustainable management of water treatment sludge through 3'R' concept | 2016 | New fuels—Sewage sludge treatment technologies, including RKs, in cement production | C | [97] |
| 16 | Gikas P. | Towards energy positive wastewater treatment plants | 2017 | New fuels—Usage of RKs within WWT technology | C | [98] |
| 17 | Chanaka Udayanga WD, Veksha A, Giannis A, Lisak G, Chang VW-C, Lim T-T. | Fate and distribution of heavy metals during thermal processing of sewage sludge | 2018 | Optimization—Heavy metal distribution and recovery within wastewater treatment (WWT) processes | P | [99] |
| 18 | Breitkopf, A. | Klärschlammentsorgung in Deutschland 2019 | 2019 | New fuels—Share of the thermal treatment of sewage sludge in Germany | C | [100] |
| 19 | Soares RB, Martins MF, Gonçalves RF. | A conceptual scenario for the use of microalgae biomass for microgeneration in wastewater treatment plants | 2019 | Optimization—Usage of RKs within microalgae production technology | C | [101] |
| 20 | Schnell M, Horst T, Quicker P. | Thermal treatment of sewage sludge in Germany: A review | 2020 | New fuels—Sewage sludge incineration technologies, including RKs | C | [102] |
| 21 | Reckter, B. | Phosphorrecycling aus Klärschlamm | 2021 | New fuels—Typical reactors for sewage sludge thermal treatment | C | [103] |

Ca—Categorization: P—process; E—energy; C—combination of both.

Due to the composition of heavy metals or pathogenic organisms, sewage sludge landfilling has been becoming more restricted over the past two decades [95]. EU member states deal with Council Directive 86/278/EEC of 12 June 1986 on the protection of the environment, and in particular of the soil, provided that the sewage sludge is used in agriculture. However, this directive is quite out-of-date and the new legislation is discussed. Nowadays, some EU countries deal with this issue on their own.

A good example is Germany, where the agricultural application of sewage sludge is strictly restricted. In 2019, almost 75% of sewage sludge production was treated thermally [100]. Fluidized beds or rotary kilns are the most common method of the mono-combustion of this waste [103].

The thermal processing of sewage sludge increases the carbon-rich organic composition while decreasing the sewage sludge volume. Sewage sludge contains thermally stable metals, such as chromium, mangan, nickel, etc., but also thermally unstable metals, such as mercury, cadmium, arsenic, lead, etc. The stable metals are less volatile at temperatures between 200 and 1100 °C. Thus, these metals are enriched in the residuals. The unstable metals are prone to volatilization. However, the volatilization of metals is influenced by sewage sludge characteristics and the type of thermal process [99]. The energy content of sewage sludge varies (roughly about 7 MJ/kg in the case of non-dried sludge and about 12 MJ/kg in the case of pre-dried sludge). Due to high temperatures in cement kilns, when pathogenic organisms and hazardous organic compounds are destructed, sewage sludge

disposal in these facilities can be recommended. The experiments reveal that the sewage sludge co-combustion within cement production processes does not exceed the mercury emission standards [90]. In the alkaline environment in the cement rotary kiln, the acidic molecules in exhaust gases are chemically bound [93].

Due to the low supply and high demand for phosphor, there are strong efforts towards its recovery. Phosphate recovery from municipal wastewater using a rotary kiln is investigated by Cohen [92]. The AshDec® process recovers phosphate from sewage sludge using a thermochemical treatment in a rotary kiln. It was successfully operated in Austria from 2005 to 2008, treating four tons of sewage sludge per hour [96].

Due to the fluidized bed reactor characteristics, sewage sludge treatment within this technology is advantageous [99]. Mininni et al. [86] reported that a lower volatilization of lead and zinc is observed during incineration in a fluidized bed incinerator compared to the results obtained from a rotary kiln. Imperfect mixing in the rotary kiln causes pyrolysis pockets that induce volatilization. The choice depends on process requirements.

Several technologies of sewage sludge combustion, including rotary kilns, are presented by Werther and Ogada [85]. The most common technologies for sewage sludge combustion are multiple hearth furnaces or a fluidized bed; however, they suffer from the energetic and economic disadvantages of sewage sludge treatment in the rotary kiln (caused mainly by poor fuel and feed mixing). Schnell et al. [102] recommend implementing the rotary kiln into waste incineration plants in order to use hot flue gas in counter-current flow with sewage sludge. A complex overview of sewage sludge treatment (not only combustion) is presented by Ahmad et al. [97].

Soares et al. [101] studied microalgae as a potential source of biomass for energy production (electricity generation) within WTPs. A rotary dryer serves for drying the biomass to meet the moisture requirements for the gasifier [101]. Due to high moisture, sewage sludge is usually pre-dried before combustion. These processes (drying, combustion) could be separated or connected. Rotary kilns are commonly used for drying sludge, biosolids, and microalgae within WTP processes [94].

In order to reduce the high energy consumption of WTPs, Gikas [98] developed a new WTP model based on advanced microsieving and filtration. In this model, the rotary kiln is placed after the auger press and reduces moisture from 55% to 20% to prepare the feed for the gasification process. Compared to the regular WTP process, the newly proposed one produces a more favorable energy balance [98].

Siemens Schwell–Brenna Technology used a rotary kiln for the pyrolysis process. Sewage sludge was mixed with crushed waste. The residuals from the kiln were subsequently combusted with gases in the boiler. Waste heat from the boiler was sent back to pre-heat the charge [93].

Mixing different fuels/waste is a promising approach, where particular strengths of fuel might increase and weaknesses might decrease. The goal is to find an optimal mixture in order to meet emission, product, and process requirements. For example, Shen and Zhang [87] compared several technologies for the pyrolysis of municipal waste mixed with sewage sludge.

Different waste material mixing is a prospective method of waste disposal. The co-combustion of sewage sludge with lignite [88], wood [91], or municipal waste [87] has been investigated. Horsley et al. [53] describe an option of waste tires, solidified sewage sludge, and animal residuals used as alternative fuels in Portland cement production.

Shen and Zhang [89] propose several recommendations in order to control the yield of the pyrolysis of putrescible garbage and sewage sludge:

- The higher the temperature and the shorter the residence time, the higher the oil yield;
- The longer the residence time, the lower the oil viscosity;
- The viscosity of oils from putrescible garbage is relatively higher than the oil from sewage sludge [89].

Just 21 of 131 articles are concerned with sewage sludge treatment. Seven articles are categorized into process engineering, while the rest are categorized into the combination of

the process and energy industries. In developed countries, the thermal treatment of sewage sludge still increases its share of the utilization/disposal of this waste. Not only due to its heavy metal content, its agricultural application is and will be more restricted. The co-combustion of this waste in the cement rotary kiln or coal-based power plants is already common. Similar to the MSW treatment, the pyrolysis pre-treatment of sewage sludge in the rotary kiln and further co-combustion in a regular power plant is advantageous and promising. Current research is also focused on material utilization, especially phosphorus recovery. This strategic commodity is crucial for agricultural sustainability.

### 4.3.3. Waste Tire Treatment

Generally, a tire is a co-polymer of long-chain polymers. Tires contain isoprene, styrene, butadiene, etc. In 2011, almost 15 million tons of tires were produced [104]. Their disposal is quite disputable; landfilling is very dangerous. The fire hazard of scrap tires also extremely burdens the environment. Recycling end-of-life tires relates to high costs, and final product parameters are questionable [105]. Pyrolysis and gasification (studied by Donatelli et al. [106]) are other methods of treatment. However, the most common method of tire disposal is thermal treatment in cement rotary kilns [104]. Relevant articles are presented and categorized in Table 7.

**Table 7.** Articles reviewed in Section 4.3.3.

| Nm. | Authors | Title | Year | Motivation—Remarks | Ca. | Cit. |
|---|---|---|---|---|---|---|
| 1 | Pasel C, Wanzl W. | Experimental investigations on reactor scale-up and optimisation of product quality in pyrolysis of shredder waste | 2003 | New fuels—Shredder waste pyrolysis | C | [107] |
| 2 | Li S-Q, Yao Q, Chi Y, Yan J-H, Cen K-F. | Pilot-Scale Pyrolysis of Scrap Tires in a Continuous Rotary Kiln Reactor | 2004 | New fuels—Pyrolysis of waste tires | C | [108] |
| 3 | Díez C, Sánchez ME, Haxaire P, Martínez O, Morán A. | Pyrolysis of tyres: A comparison of the results from a fixed-bed laboratory reactor and a pilot plant (rotatory reactor) | 2005 | New fuels—Pyrolysis of waste tires—comparison of a fixed bed and rotary kiln | C | [109] |
| 4 | Donatelli A, Iovane P, Molino A. | High energy syngas production by waste tires steam gasification in a rotary kiln pilot plant | 2010 | New fuels—Steam gasification of waste tires within an RK | C | [106] |
| 5 | Llompart M, Sanchez-Prado L, Pablo Lamas J, Garcia-Jares C, Roca E, Dagnac T. | Hazardous organic chemicals in rubber recycled tire playgrounds and pavers | 2013 | Optimization—Material utilization of waste tires | P | [105] |
| 6 | Hossain AK, Davies PA. | Pyrolysis liquids and gases as alternative fuels in internal combustion engines—A review | 2013 | New fuels—Pyrolysis of waste tires | C | [110] |
| 7 | Martínez JD, Puy N, Murillo R, García T, Navarro MV, Mastral AM. | Waste tyre pyrolysis—A review | 2013 | New fuels—Pyrolysis of waste tires | C | [111] |
| 8 | Williams PT. | Pyrolysis of waste tyres: A review | 2013 | New fuels—Pyrolysis of waste tires | C | [112] |
| 9 | Lamas W de Q, Palau JCF, Camargo JR de. | Waste materials co-processing in cement industry: Ecological efficiency of waste reuse. | 2013 | New fuels—Waste tire usage within cement production | C | [113] |
| 10 | Antoniou N, Zabaniotou A. | Features of an efficient and environmentally attractive used tyres pyrolysis with energy and material recovery | 2013 | New fuels—Pyrolysis of waste tires—comparison of reactor types | C | [114] |
| 11 | Acevedo B, Barriocanal C, Alvarez R. | Pyrolysis of blends of coal and tyre wastes in a fixed bed reactor and a rotary oven | 2013 | New fuels—Pyrolysis of waste tires—comparison of a fixed bed and rotary kiln | C | [104] |

**Table 7.** *Cont.*

| Nm. | Authors | Title | Year | Motivation—Remarks | Ca. | Cit. |
|---|---|---|---|---|---|---|
| 12 | Antoniou N, Stavropoulos G, Zabaniotou A. | Activation of end-of-life tires pyrolytic char for enhancing viability of pyrolysis—Critical review, analysis and recommendations for a hybrid dual system | 2014 | New fuels—Pyrolysis of waste tires | C | [115] |
| 13 | Hita I, Arabiourrutia M, Olazar M, Bilbao J, Arandes JM, Castaño P. | Opportunities and barriers for producing high quality fuels from the pyrolysis of scrap tires | 2016 | New fuels—Pyrolysis of waste tires | C | [116] |
| 14 | Ayanoğlu A, Yumrutaş R. | Rotary kiln and batch pyrolysis of waste tire to produce gasoline and diesel like fuels | 2016 | New fuels—Pyrolysis of waste tires—pyrolytic oil usage within diesel engines | C | [117] |
| 15 | Kumaravel ST, Murugesan A, Kumaravel A. | Tyre pyrolysis oil as an alternative fuel for diesel engines—A review | 2016 | New fuels—Pyrolysis of waste tires—pyrolytic oil used for diesel engines | C | [118] |
| 16 | Horsley C, Emmert MH, Sakulich A. | Influence of alternative fuels on trace element content of ordinary Portland cement | 2016 | New fuels—Waste-derived fuels in cement production | P | [53] |
| 17 | Machin EB, Pedroso DT, de Carvalho JA. | Energetic valorization of waste tires | 2017 | New fuels—Comparison of waste tire energetic utilization approaches | E | [119] |
| 18 | Lewandowski WM, Januszewicz K, Kosakowski W. | Efficiency and proportions of waste tyre pyrolysis products depending on the reactor type—A review | 2019 | New fuels—Pyrolysis of waste tires—comparison of reactor types | C | [120] |

Ca—Categorization: P—process; E—energy; C—combination of both.

The combustion of waste tires in the rotary kiln shows low operating costs. On the other hand, it requires complex technologies for flue gas treatment (such as particulate filtration and emission controlling) [119]. The advantage of waste tires used as fuel is also relatively low adaptation costs [116].

Waste tires could be pyrolyzed to produce char, oil, and gas [110–112]. Combustion within cement production is advantageous due to the high energy content of waste tires (about 31.4 MJ/kg) [113]. As mentioned in the previous section, waste mixing and combustion or pyrolysis have a strong potential in the waste-to-energy approach. Waste tires mixed with solidified sewage sludge and animal residuals for utilization within Portland cement production have been studied by Horsley et al. [53]. The authors state that the amount of alternative fuels treated within cement production is limited. The most limiting factor is increased ash production caused by alternative fuel co-incineration. Due to the wide range of potential wastes, the share of alternative fuels within cement production will increase in the future. However, the clinker quality must be controlled.

The pyrolysis of waste tires is usually carried out in indirectly heated rotary kilns. The advantages are the stability of process conditions and the quality of mechanical treatment in the kiln [107]. The final pyrolytic product could be activated charcoal. Antoniou et al. [115] reviewed the pyrolysis of waste tires in order to find optimal process conditions. The quality of the final product and the influence of process parameters on the final product quality have also been studied.

Several technologies of waste tire pyrolysis, including the treatment in the rotary kiln, have been studied and compared [114,120]. The conversion of waste tires into wax, heavy and light oil, carbon black, and non-condensable gas in the rotary kiln has been studied by Acevedo et al. [104] and Ayanoglu and Yumrutas [117].

Two experiments of waste tire pyrolysis, one in the rotary kiln reactor and the other in the fixed bed reactor, were carried out by Diez et al. [109] to compare two different technologies for waste tire treatment. The amount of treated waste (the particle size is between 10 and 50 mm) was 20 kg, and the process temperature was 550 °C. Due to different heating ramps and residence times, different final product compositions and heating values were obtained. However, the differences were not significant.

A continuous rotary kiln reactor for the pyrolysis of waste tires was used in the experiment carried out by Li et al. [108]. The high heating values of yielded oils are approximately 40 MJ/kg; the original pyrolytic char meets all requirements to be used as an adsorbent [108].

There is intense pressure on the use of alternative fuel in combustion engines. Compared to biofuels produced from, e.g., rapeseed, the waste tires for the production of pyrolysis oil do not occupy the fields where food crops are grown. This can solve the problem of treating waste to produce a valuable product. Kumaravel et al. [118] believe that the tire pyrolysis oil used in diesel engines is worthwhile.

Just 18 of 131 articles are concerned with waste tire utilization; most of them are categorized into a combination of the process and energy industries. Due to a relatively stable and high energy content, waste tires are mostly co-combusted in the cement rotary kiln without any significant negative impact on the clinker quality. Fourteen articles deal with the pyrolysis or gasification of waste tires. Pyrolytic oil production with the aim of replacing at least part of standard fossil fuel consumption in diesel engines is promising. Thus, rotary kilns represent a key technology in waste tire utilization.

### 4.4. Computational Modeling, Controlling, and Artificial Intelligence

The construction of RKs is well-known and relatively simple. However, several subsequent processes that appear during its operation are not easy to describe, i.e., modeling or controlling. Every process requires specific temperature conditions and specific residence time within every part of the process. Therefore, numerical modeling or artificial intelligence could be very beneficial. Relevant articles are presented in Table 8.

**Table 8.** Articles reviewed in Section 4.4.

| Nm. | Authors | Title | Year | Motivation—Remarks | Cit. |
|---|---|---|---|---|---|
| 1 | Suzuki K, Iriyama M, Nakamori Y. | Statistical analysis of dynamics of a rotary kiln sewage sludge incinerator using fuzzy modeling | 1992 | Optimization—Analysis and modeling of RK dynamic performance | [121] |
| 2 | Cho WS, Roh SD, Kim SW, Jang WH, Shon SS. | The process modeling and simulations for the fault diagnosis of rotary kiln incineration process | 1998 | Optimization—Control of an RK using an online system | [122] |
| 3 | Kalogirou SA. | Artificial intelligence for the modeling and control of combustion processes: a review | 2003 | Optimization—Review of AI in the modeling and control of combustion processes | [123] |
| 4 | Finnie GJ, Kruyt NP, Ye M, Zeilstra C, Kuipers JAM. | Longitudinal and transverse mixing in rotary kilns: A discrete element method approach | 2005 | Optimization—Mixing modeling with the use of a discrete element method | [4] |
| 5 | Nielsen, A. R., Larsen, M. B., Dam-Johansen, K., Glarborg, P. | Combustion of large solid fuels in cement rotary kilns | 2012 | New fuels—Alternative fuels in cement production | [2] |
| 6 | Li T, Zhang Z, Chen H. | Predicting the combustion state of rotary kilns using a Convolutional Recurrent Neural Network | 2019 | Optimization—Prediction of combustion using the neural network | [124] |
| 7 | Chang, T.-B., Lee, C.-Y., Ko, M.-S., Lim, C.-F. | CFD simulations of rotary BOF slag carbonation kiln reactor with cyclone flow | 2020 | Numerical analyses for Carbon Capture Technology optimization | [125] |
| 8 | Wang M, Chen E, Liu P, Guo W. | Multivariable nonlinear predictive control of a clinker sintering system at different working states by combining artificial neural network and autoregressive exogenous | 2020 | Optimization—Control of an RK using an extreme learning machine-autoregressive exogenous model | [126] |
| 9 | Ko, M.-S., Chang, T.-B., Lee, C.-Y., Huang, J.-W., Lim, C.-F. | Optimization of Cyclone-Type Rotary Kiln Reactor for Carbonation of BOF Slag | 2021 | Numerical analyses for Carbon Capture Technology optimization | [127] |

Due to the unstable composition of alternative fuels, their devolatilization is a complex task that can influence the quality of the process (clinker or lime production). Moreover, devolatilization and further oxidization are influenced not only by the rotational speed/kiln inclination or the process temperature but also by the particle size or shape, fill ratio, bed oxygen concentration, etc. Thus, modeling these processes is quite a complex and essential research method [2].

In 1992, Suzuki et al. [121] studied the modeling and analysis of the dynamic performance of the rotary kiln incinerator. The process in the rotary kiln is sewage sludge incineration, where a large number of input disturbances appear. In 1998, Cho et al. [122] used an online system called RKEXPERT to supervise the processes in the rotary kiln. They focused on fault diagnosis and feedback. They completed an intelligent system to provide integrity in optimal control.

The cement rotary kiln is usually controlled manually by experienced operators. However, due to many nonlinearities within the process, a precise estimation of controlled variables cannot be provided.

Numerical analyses using Computational Fluid Dynamics (CFD) also play an important role in Carbon Capture Technologies. An example of that is an optimized model of a Cyclone-Type Rotary Kiln Reactor. This approach is beneficial due to how it stores carbon dioxide and also stabilizes the slag (product of steel-making processes) that can be reused as a construction material afterwards [125]. The optimized model increases the carbon dioxide storage capacity in slag [127].

Artificial intelligence (AI) systems can learn from examples. They are fault-tolerant (capable of operating with noisy and incomplete data) and can perform predictions. Thus, it is beneficial to use AI in rotary kiln control systems. Wang et al. [126] used the extreme learning machine-autoregressive exogenous model to describe the nonlinearities within the process. The model can precisely control the process performance. Moreover, it can decrease the $NO_x$ emission concentration. Kalogirou [123] reviewed the role of AI in modeling and predicting the performance and control of combustion processes in general.

Li et al. [124] used a neural network to predict the combustion state in the rotary kiln. The neural network analyzed the sequences of flame images. The proposed method was tested on real data. The prediction accuracy reached more than 93%; therefore, a high potential for industrial applications can be expected.

A discrete element method could be used for longitudinal and transverse mixing modeling in rotary kilns. Finnie et al. [4] focused on the main operating conditions, such as the fill ratio and rotational speed, and quantitative longitudinal and transverse mixing measures. It was found that the diffusion coefficient increased linearly with the rotational speed. The influence of the fill ratio on the diffusion coefficient was relatively small. The mixing speed decreases with increasing rotational speed and increasing fill ratio [4].

Section 4.1 points out that the energy requirements of cement and lime production are very high and so is the potential for its optimization. In this field, the research is focused on particle motion modeling, operation characteristics (e.g., fill ratio) relationships, control optimization, or emission reduction. The aim of the research represented by the nine articles in Table 8 is the optimization of the processes to reach efficiency maximization and negative environmental impact mitigation. Due to the very involved processes that appear in RK, the modeling and AI application is a complex task. However, the potential of numerical analysis or artificial intelligence is very promising.

There are several other applications of RKs that are not discussed in the individual sections. An example could be the extraction and separation of manganese from specific ores. The review of these processes, where the ore is first crushed and then continues to the rotary kiln for carbothermal reduction roasting, is presented by Liu et al. [128]. Another example is the RK used in the aluminum industry, which has a significant negative environmental impact. The whole aluminum production process, with a focus on its environmental impact, was examined by Brough and Jouhara [129]. A wide range of waste heat recovery systems was discussed, and an optimal solution was recommended. The

authors concluded that the secondary methods (recycling) of aluminum production are more efficient than the production from primary sources.

The iron ore pellet production within coal-fired RKs can be advantageous for the environmental protection and energy conservation of the whole ironmaking process. However, deposit formation can substantially decrease the efficiency of its production. Wang et al. [130] reviewed the deposit formation mechanism to ensure the efficient production of the pellets. The authors recommend increasing the compression strength of preheated pellets, choosing high-quality raw materials with low alkali metals, or improving the combustion of pulverized coal.

Electric arc furnace dust (EAFD) is a kind of metallurgical solid waste. A complex overview of its material utilization is presented by Wang et al. [131]. Rotary kilns take place within the recovery process of valuable metals such as zinc, lead, or iron from EAFD. The authors propose a new process for treating EAFD. The zinc oxide micropowder and high-valued Fe–Ni–Cr alloy preparing for development by a pyrometallurgical process of electric arc furnace dust is prospective.

## 5. Discussion

Even though the construction of a rotary kiln is quite simple, there is still ongoing, fundamental, and applied research in this area, and many questions have not been answered yet.

The highest number of 76 articles are classified into the "Combination" category and 29 into the "Process" category. Only six analyzed articles are connected with a pure energy point of view. A combination of process and energy engineering seems to be the most significant trend in rotary kiln research. The purpose of this process approach is to make the system more efficient and effective. To reach that, the processes and their interactions must be well understood. The benefits of this approach in terms of rotary kiln operation are energy/emission savings, the utilization of renewable fuels, the better material/energy utilization of various wastes, etc.

The potential for alternative fuels to replace fossil fuels in clinker production is estimated to be about 40%. This means an approximate $CO_2$ production of 0.53 tons per ton of clinker instead of 0.85 ton of $CO_2$ using nonrenewable fuels. However, the final product quality may change and must be controlled due to the ash formation, clogging, or mercury content in the clinker [56]. Many RKs of older types are not operated according to new findings and standards. The investment to minimize the negative impact of production should be considered. Due to high energy demands, the payback period of these investments is relatively low.

Since the processes in the RK are quite complex, artificial intelligence could be used in order to optimize its operation. The potential for computational analysis, modeling, or controlling is very promising.

MSW is mostly incinerated in moving grate incinerators due to the ability to treat large volumes without pre-treatment, such as sorting or shredding [52]. Rotary kilns used within waste material energy or material utilization are an advantageous solution for hazardous waste due to the processability of any type of waste (solid, liquid, or gaseous) and higher process temperatures. Rotary kilns are also suitable for the processing of large volumes of generated waste.

In order to decrease the amount of landfilled waste, co-combustion in incineration plants is highly recommended, provided that the network of plants for thermal waste treatment is dense, distribution and transport are faster, cheaper, and more accessible, and the overall $CO_2$ production is lower. However, MSW combustion in the absence of pre-treatment in regular incineration plants causes problems due to the unstable content, increased fouling, slagging, and corrosion. This could be solved by MSW pre-treatment using pyrolysis in the rotary kiln or by sorting and mixing MSW and industrial waste to produce refuse-derived fuels (RDF). MSW pyrolysis treatment is very promising, especially

if the waste heat from the incineration plant, biogas power plant, or WtE plant (e.g., in summer) is used for pyrolysis heating.

The combustion of municipal solid waste within regular coal-based power plants is disputable due to its unstable content. For example, chlorine content is responsible for corrosion. Thus, the recycled MSW or industrial waste could be crushed and mixed to produce refuse-derived fuel RDF. This certified (the certificate ensures the quality of the fuel—the chlorine content is usually strictly controlled) fuel could be co-incinerated in the cement kiln (21 papers are focused on alternative fuel combustion within cement and lime production) or incineration plants at a low percentage without any negative consequences. As mentioned in the previous section, this is crucial in countries with a high number of incineration plants and a low number of WtE plants. For example, there are 26 incineration plants in the Czech Republic, but only four WtE plants.

Auspicious is the idea of utilizing the waste heat from the plant for digestate (byproduct from biogas power plant processes) treatment in the rotary kiln. This combined process has the potential to increase the electricity production of the biogas power plant.

Due to the amount of sewage sludge that is still increasing, there is a strong effort in material and energy utilization. The agricultural application is and will be more restricted not only due to heavy metal content. In developed countries, the thermal treatment of sewage sludge still increases its share of the utilization/disposal of this waste [95]. Its mono-combustion usually takes place in a fluidized bed or a rotary kiln. The co-combustion of sewage sludge in cement rotary kilns or incineration plants is already common. Current research is also focused on material utilization, especially phosphorus recovery. This strategic commodity is crucial for agricultural sustainability.

Also, biomass combustion causes several problems. Ash composition during biomass incineration varies according to the biomass type, which increases the fouling, slagging, and corrosion responsible for lower process efficiency and higher maintenance costs. Moreover, untreated biomass usually has a high moisture content that causes a low energy content and leads to high storage costs. Thus, biomass pre-treatment in the rotary kiln, such as torrefaction, drying, gasification, pyrolysis, etc., seems to be advantageous and worth studying. Finding an optimal mixture of fuel/waste is a promising approach, where particular strengths of fuel might increase and weaknesses might decrease. The goal is to find a fuel mixture suitable to meet emission, product, and process requirements.

Cement kilns are commonly used for waste tire co-combustion. Recycling this waste is expensive, and the product quality is questionable. Thus, thermal treatment is a suitable method of utilization [116]. Combustion within cement production is advantageous due to the high energy content of waste tires (ca. 31.4 MJ/kg) [123]. There is also intense pressure on alternative fuels to replace at least a part of standard fossil fuel consumption in diesel engines. Compared to biofuels produced, e.g., from rapeseed, the waste tires used for pyrolysis oil production do not occupy agricultural fields that could be used to grow food crops. The use of waste tire pyrolysis oil in diesel engines is a suitable option [131]. Rotary kilns represent a key technology in waste tire utilization.

A new approach to RK research is based on solar energy utilization and storage. Sixteen articles are focused on this phenomenon. Even though RKs are not the only facility studied within this research, the above RK processes are commonly investigated. Except for the sensible and latent type of heat storage, the thermo-chemical technology has probably the highest potential due to its ability to store energy at ambient temperature, long-term stability, and high energy density. However, there is still a significant difference in heat stored in some feasible solar energy carriers and common fossil fuels such as lignite (15 MJ/kg vs. 400 kJ/kg). Hydrogen production is also an auspicious technology of solar energy utilization within the rotary kiln process, as reported by Grassmann et al. [39].

Due to the high energy demand of lime or clinker production, the use of solar energy in this industry is advantageous. For commercialization, simpler processes with a single reaction, such as upgrading carbon feed or the production of industrial commodities, are recommended [37]. Moreover, the processes of solar methane reforming or lime production

are more efficient in comparison with others. It is expected that solar energy utilization systems will be able to cover at least a part of fossil fuel consumption in the future.

The most significant trend in RK research is sustainability and efficiency maximization, where the border between the process and energy application is being eliminated. The RK plays both roles: it has an irreplaceable role in building material production, such as clinker and lime production. Moreover, industrial waste (e.g., automotive shredder residues), waste tires, waste oil, sewage sludge, plastics, etc., are combusted within this production. Thus, the rotary kiln is responsible for a significant amount of waste utilization and disposal.

Concentrated solar power within solar energy utilization seems to be a new, auspicious approach to green energy production, where rotary kilns play an essential role, especially in thermochemical heat storage systems.

## 6. Conclusions

Let us offer the answers to the questions defined earlier in this review paper:

What are the main rotary kiln research areas?

Based on the state of the art presented in this review paper, the main RK research areas seem to be the waste-to-energy approach, the pyrolysis of various types of waste, the optimization of cement and lime production, or solar energy utilization. In clinker and lime production, the research is focused on $CO_2$ emission mitigation, energy recovery, or loss mitigation and the use of alternative fuels. Approximately half of the articles cited in Section 4.2 deal with heat energy storage. The rest deal with solar energy utilization within RK operation, such as pyrolysis or lime and clinker production. The main research areas in WtE research connected with RK represent a search for various wastes suitable for combustion/pyrolysis within RK processes, the optimization of RK operation, and negative environmental impact mitigation.

What are the main motivations for research on rotary kilns?

The main motivation for RK research is connected with the high energy requirements of this facility. Departure from fossil fuels leads to an increase in the use of renewable fuels or the utilization of waste. Restrictions for waste landfilling also lead to a higher rate of recycling and energy utilization. Another motivation is to utilize and store clean and cheap solar energy, replace standard fossil fuels, and minimize the negative environmental impact of RK operation. Also advantageous is the search for an optimal waste mixture for combustion/pyrolysis. Pyrolysis output products, such as gas, oil, or biochar, could be used as fuel and replace standard fossil fuels. Waste distribution into WtE plants or other suitable facilities also seems to be worth studying. Thus, the more plants that can incinerate the waste, the easier the distribution, resulting in lower transport costs and lower emissions.

Agricultural applications of sewage sludge become more restricted, and not only due to the heavy metal content that can pollute underground waters. Thus, the thermal treatment of this material includes the RK processes. Current research is also focused on the material utilization of sewage sludge, especially phosphorus recovery. Also advantageous is pyrolytic oil production to replace at least a part of standard fossil fuel consumption in diesel engines.

What are the ways towards energy sustainability in rotary kiln operation?

The long-term energy sustainability of RK operation leads to better heat utilization systems, the use of alternative fuels, or new technologies for energy utilization, such as solar system integration. As the result, there is the maximum responsible use of the energy produced by RK, whether it is the main purpose of RK (energetics) or waste heat (process engineering). According to the author's point of view, more alternative fuels and waste should be treated in local and accessible plants (WtE plants and incineration plants) in the future. To ensure the effective operation of these processes, part of the waste/fuel will have to be pre-treated; in this respect, the rotary kilns play a key role.

So far, any paper focused on the influence of alternative fuel composition on particulate matter formation has been published. Finding a good mix of alternative fuels (such as sewage sludge, municipal solid waste, biomass, etc.) can optimize fouling and slagging within rotary kiln processes. This seems to be a research gap that should direct future research. Publications on sub-topics predominate. Therefore, the authors recommend focusing future research on integrated solutions using a process systems approach. These complex solutions always carefully consider both the operational (mass calculations) and the energy side (energy calculations) of the integration, which is the basis for the long-term sustainability of environmental protection.

**Author Contributions:** Conceptualization, J.B. and V.M.; methodology, J.B. and V.M.; validation, P.S., I.H., J.H.; formal analysis, J.B., I.H.; investigation, J.B., P.S.; resources, J.B. and J.H.; writing—original draft preparation, J.B.; writing—review and editing, V.M.; visualization, I.H.; supervision, J.H. and P.S.; project administration, P.S.; funding acquisition, V.M. All authors have read and agreed to the published version of the manuscript.

**Funding:** This research was funded by Ministry of Education, Youth, and Sports of the Czech Republic under OP RDE (grant number CZ.02.1.01/0.0/0.0/16_026/0008413) "Strategic Partnership for Environmental Technologies and Energy Production".

**Institutional Review Board Statement:** Not applicable.

**Informed Consent Statement:** Not applicable.

**Data Availability Statement:** Not applicable.

**Conflicts of Interest:** The authors declare no conflict of interest. The funders had no role in the design of the study; in the collection, analyses, or interpretation of data; in the writing of the manuscript, or in the decision to publish the results.

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
