# Peer review of "Rotary Kiln, a Unit on the Border of the Process and Energy Industry—Current State and Perspectives"

_sustainability, doi:10.3390/su142113903_

Round 1
Reviewer 1 Report
Reviewer’s report on review manuscript sustainability-1906350
The authors attempted to assess a very wide and complex topic of recent research in the field of rotary kilns both from process- and energy point of view. There is no doubt that a thorough review on recent advances in this field can produce relevant findings and conclusions, useful for both academicians and practitioners. It can be stated that the authors succeeded – partly. Among the positives of the manuscript, the following can be mentioned:
- authors are experienced and this is seen throughout the manuscript, starting from definition of research questions, organization of cited papers into relevant categories and point by point discussion regarding individual papers
- manuscript contains a lot of interesting information
Unfortunately, the authors should improve in academic writing, in the way how to prepare data for a proper review and how to prepare manuscript for peer review. My objections are related to:
- writing style which mixes up past and present tense quite often; the manuscript would certainly benefit from careful language editing
- Method part deserves significant improvement, lacking key information and justifications; taking ScienceDirect as the primary platform for source papers means neglecting tons of relevant research published elsewhere
- Most recent papers (2021 and 2022) are not covered by this manuscript, negatively affecting the sense of its actuality
- Manuscript preparation should be more careful, including proper pdf version proofreading before its submission to the journal as to avoid confusion in hypertext links, reducing the manuscript legibility
- Results and Conclusions section should be reorganized, as Conclusions is more “Discussion”-like that a brief two- or three-paragraph long item containing the distillate (essence) of the review findings.
Despite those issues, manuscript is relevant for Sustainability journal and a thorough revision, if following my recommendations closely, should improve its quality and its sense of actuality sufficiently. Authors are encouraged to review their manuscript; for this purpose a list of individual queries is attached:
Abstract, line 13 and 14: Please reconsider using past tense in this sentence; you use present tense in abstract so this one sentence should be harmonized with the rest of abstract.
Introduction:
Line 35: it is the first figure you refer to, why is it termed Figure 2? Both reference to Figure 1 as well as the figure 1 are missing in the manuscript.
Line 40 and 41: This is an example of popular rather than scientific writing. Please try to be more specific and avoid using unspecific expressions like “great amount of energy” throughout the manuscript.
Line 43: Referring to “Chapter 0” – check. See lines 60 and 67 as well as the rest of the manuscript – obviously the hypertext connections do not work in your manuscript. This is a frequent issue and careful pdf proofreading before its submission should be performed to avoid this. It worsens the legibility of the manuscript significantly and would normally be a reason for paper rejection at initial evaluation stage!
Methodology:
Justification for choosing ScienceDirect as the source for research papers should be provided.
I recommend using SCOPUS due to its much wider coverage of journals by various publishing houses. By relying on ScienceDirect only, the authors effectively omit any relevant research and review papers published in journals by reputed publishers such as Springer, Taylor and Francis and in multitudes of indexed open access journals. Though not as formal and strict as a systematic review paper, a semi-systematic review should use and work multiple sources as well. Then, it seems awkward that with such literature scope the authors intend to publish their review paper in a MDPI journal, while not including a single reference to a paper published in this publishing house.
Justification for choosing review paper only for further analysis should be provided.
Information when the searching process was initiated is missing. I lack the newest papers in your manuscript, see the next comment as well.
Out of several tens of papers listed and assessed in Table 1-4 and 6-8, only one was published in 2021; the rest were published in 2020 and earlier. It seems that most recent papers published in 2021 and 2022 are not included. The actuality of this manuscript is thus questionable.
Results:
Line 180: I did not see anywhere in the previous text reference to 500 articles. Method chapter stated to be working with 400 articles initially.
Lines 234-252 Please use relative expressions (% of…) rather than absolute numbers. This helps the reader to assess the significance of proposed measures.
Line 434-435: To state a single CO2 emission rate representing the whole spectrum of fossil fuels fired in rotary kilns is inappropriate. Be more specific to which fuel the given figure belongs.
Line 586: Check and use “.” as decimal separator instead of “,”. Check the whole manuscript.
Line 667-668: Incomplete sentence.
This manuscript chapter should be concluded by an executive summary rather than writing a 3 page-long Conclusions.
Conclusions: Standard way of writing Conclusions includes summarizing the most important findings and proposing new research directions. No new information should appear in this section. Yet my feeling is that the authors exceeded this scope and wrote Conclusions in form of Discussion which is inappropriate. Text passages in lines 910-929 and 941-949 contain new information that is not sufficiently reflected in previous chapters.
General writing style: The authors mix up past and present tense which does neither contribute to manuscript clarity nor its legibility. See lines 339-344 as example with use of mixed-up tenses. You should pay close attention to this aspect throughout the manuscript.
Author Response
Abstract, line 13 and 14: Please reconsider using past tense in this sentence; you use present tense in abstract so this one sentence should be harmonized with the rest of abstract.
The tense was changed. Thank you.
Introduction:
Line 35: it is the first figure you refer to, why is it termed Figure 2? Both reference to Figure 1 as well as the figure 1 are missing in the manuscript.
Thank you very much for this point. Now, all figures are termed continuously as well as their references.
Line 40 and 41: This is an example of popular rather than scientific writing. Please try to be more specific and avoid using unspecific expressions like “great amount of energy” throughout the manuscript.
The sentence was rewritten to more scientific writing as” RK are usually very energy demanding”. At this point, the specific calculation is not used. The authors aim to highlight the RK energy demands in general.
Line 43: Referring to “Chapter 0” – check. See lines 60 and 67 as well as the rest of the manuscript – obviously the hypertext connections do not work in your manuscript. This is a frequent issue and careful pdf proofreading before its submission should be performed to avoid this. It worsens the legibility of the manuscript significantly and would normally be a reason for paper rejection at initial evaluation stage!
Thank you very much for your point. All references were checked and improved where needed.
Methodology:
Justification for choosing ScienceDirect as the source for research papers should be provided.
I recommend using SCOPUS due to its much wider coverage of journals by various publishing houses. By relying on ScienceDirect only, the authors effectively omit any relevant research and review papers published in journals by reputed publishers such as Springer, Taylor and Francis and in multitudes of indexed open access journals. Though not as formal and strict as a systematic review paper, a semi-systematic review should use and work multiple sources as well. Then it seems awkward that with such literature sources as well. Then, it seems awkward that with such literature scope the authors intend to publish their review paper in a MDPI journal, while not including a single reference to a paper published in this publishing house.
You are right. Scopus is a wider database than ScienceDirect. Since we studied the primary sources in our initial database, our article include also papers out of the ScienceDirect database. However, the database of ScienceDirect includes more than 400 review articles mentioning Rotary kilns. It took us almost a year to study these papers in deep. Thus, we think ScienceDirect overlaps a significant and sufficient research field for a complex overview.
Nevertheless, in this revision, our paper was completed by several new articles from the Scopus database including MDPI journal (see references 22, 23, 128, and 131). These papers have confirmed the conclusion of our paper.
Justification for choosing review paper only for further analysis should be provided.
Information when the searching process was initiated is missing. I lack the newest papers in your manuscript, see the next comment as well. Out of several tens of papers listed and assessed in Table 1-4 and 6-8, only one was published in 2021; the rest were published in 2020 and earlier. It seems that most recent papers published in 2021 and 2022 are not included. The actuality of this manuscript is thus questionable.
Since the analysis of the initial database created in 2021 took us a year, the newest papers published in 2022 were missing. Therefore our database was enriched by the newest review articles from the Scopus database (see references 22, 23, 128, and 131).
Results:
Line 180: I did not see anywhere in the previous text reference to 500 articles. Method chapter stated to be working with 400 articles initially.
The mistake was found and solved. Thank you.
Lines 234-252 Please use relative expressions (% of…) rather than absolute numbers. This helps the reader to assess the significance of proposed measures.
Thank you very much. It was improved.
Line 434-435: To state a single CO2 emission rate representing the whole spectrum of fossil fuels fired in rotary kilns is inappropriate. Be more specific to which fuel the given figure belongs.
Thank you. The fuel specification was added.
Line 586: Check and use “.” as decimal separator instead of “,”.Check the whole manuscript.
The mistake was found and solved. Also, the whole document was checked. Thank you.
Line 667-668: Incomplete sentence.
If we are not mistaken, it is the part: “Sewage sludge is a mud-like byproduct of this process. This waste in the EU is estimated at 11.25 million tons, while half is used in agricultural applications [41]. Developed countries produce more sewage sludge than developing ones.” These sentences seem complete to us.
This manuscript chapter should be concluded by an executive summary rather than writing a 3 page-long Conclusions.
Conclusions: Standard way of writing Conclusions includes summarizing the most important findings and proposing new research directions. No new information should appear in this section. Yet my feeling is that the authors exceeded this scope and wrote Conclusions in form of Discussion which is inappropriate. Text passages in lines 910-929 and 941-949 contain new information that is not sufficiently reflected in previous chapters.
Thank you for your recommendation. In this revision, our previous conclusion was split into discussion and new shorter conclusion.
General writing style: The authors mix up past and present tense which does neither contribute to manuscript clarity nor its legibility. See lines 339-344 as example with use of mixed-up tenses. You should pay close attention to this aspect throughout the manuscript.
Thank you very much for your point. The tenses were checked and united.
Reviewer 2 Report
The paper is an eclectic collection of bibliographic rather than serious scientfic research. There is no actual statement of the problem related with RK. the characterzaton of the research given by athors themeselves as semi-systemic s ethcally questionable. The artcle neglects mportant metallurgical applications of RK at all.
Authors should clearly explain what is the "boundary" between energy and process engineering and why it is represented by this aggregate????
Author Response
The paper is an eclectic collection of bibliographic rather than serious scientfic research. There is no actual statement of the problem related with RK. the characterzaton of the research given by athors themeselves as semi-systemic s ethcally questionable. The artcle neglects mportant metallurgical applications of RK at all.
Thank you for this comment. We were also surprised by the amount of literature and the importance of this topic. Thus, the result seems as a bit of bibliographical work as you point out. However, several results and recommendations are proposed in the paper. Thus, we believe these results are beneficial for the readers of the Sustainability journal.
Metallurgy is a significant part of rotary kiln application. In Scopus databases, there are approximately 160 articles mentioning RK and metallurgy. You are right, in our paper, this part of rotary kiln research plays rather a minor role. However, Figure 2 in our article mentions magnesite as a part of mineral resource processing. Also, reference 126 deals with the extraction and separation of manganese from specific ores using a rotary kiln. Reference 127 deals with aluminum production and recycling focusing on its negative environmental impact. Moreover, in this revision, several articles including that with metallurgical applications were studied and completed in our paper (see references 128 and 131). These articles confirmed the results of our paper.
Authors should clearly explain what is the "boundary" between energy and process engineering and why it is represented by this aggregate????
It is a very good question. We aim to show the overlap of the Process and Energy industry within RK applications. In several applications a charge is processed while the waste heat is subsequently utilized. We consider our approach to be appropriate to describe the current State of the Art. Both, the process itself and its energy efficiency must be taken into account.
Reviewer 3 Report
This article reviews the literature of rotary kiln and gives a historical and theoretical background, introduction to the current state of research and it also aims to find the connection between two different areas of application – process industry and energy sector. This article also shows the main rotary kiln research areas, the main motivations for research of rotary kilns and the ways towards energy sustainability of rotary kilns operation. This study is well conducted, and clearly demonstrates. The report is compact, accurate and well written for the expected audience. It is also very well oriented to technology, fitting well in the scope of the journal. This reviewer only has one suggestion. The paper does not review the simulation and optimization of Rotary Kiln Reactor using computational fluid dynamics (CFD). It is recommended to refer to the following recent papers to give readers a better understanding of CFD simulations on rotary kiln : https://doi.org/10.3390/su132011556 and https://doi.org/10.1177/0954408919883082. Moreover, the words in figure 2 are not clear, please improve them. If appropriate modifications can be made, I would to recommend the paper for publication in the Sustainability.
Author Response
This article reviews the literature of rotary kiln and gives a historical and theoretical background, introduction to the current state of research and it also aims to find the connection between two different areas of application – process industry and energy sector. This article also shows the main rotary kiln research areas, the main motivations for research of rotary kilns and the ways towards energy sustainability of rotary kilns operation. This study is well conducted, and clearly demonstrates. The report is compact, accurate and well written for the expected audience. It is also very well oriented to technology, fitting well in the scope of the journal. This reviewer only has one suggestion. The paper does not review the simulation and optimization of Rotary Kiln Reactor using computational fluid dynamics (CFD). It is recommended to refer to the following recent papers to give readers a better understanding of CFD simulations on rotary kiln: https://doi.org/10.3390/su132011556 and https://doi.org/10.1177/0954408919883082 .
Thank you very much for your suggestion. Both papers were studied and added to the paper.
Moreover, the words in figure 2 are not clear, please improve them. If appropriate modifications can be made, I would to recommend the paper for publication in the Sustainability.
Thank you very much for your recommendation. Figure 2 was improved.
Round 2
Reviewer 1 Report
Reviewer´s comments on revised review manuscript sustainability 1906350
Authors improved the manuscript from content point of view and most of my comments related to this aspect were answered / implemented in the manuscript to my satisfaction. Unfortunately, the authors did not perform a careful proofreading, causing lost of formal and formatting issues to remain in the manuscript. The list below is just an excerpt. As the authors did not follow the reviewers´ recommendations, another revision round is necessary.
The authors should focus on:
- careful reading, especially that of the newly added text
- language revision of the whole manuscript is a must
The changes to be performed are so extensive, that I believe a simple minor revision will not suffice. Thus, another major revision round is proposed, to improve the manuscript formatting and attractivity for potential audience.
List of issues (excerpt):
Several formal issues, to be dealt with in subsequent major revision, remained in the manuscript:
- formatting: text still contains some hypertext links causing f.e. reference to chapter 0 and subchapter 0 to appear in the text (line 858); similarly, check the whole Methodology part for correct referring to manuscript parts. Kindly remove any remaining links from the revised manuscript (please read the instructions for authors carefully).
- language: language polishing is necessary. Several sentences have a non-standard syntax. Especially the newly added text parts should be revised in this sense + grammar.
- references: further format unification is necessary, see refs. no. 52,18,2 as examples. Decide, whether to state full journal names, or their abbreviations, but do not mix this up.
- line 560: The sentence is to start with a reference to some artwork (probably to Table 5), but the reference is not there.
- Line 652-626: Stating a range of values implies that lower and upper limit are expressed in the same physical units (choose either MJ/kg or MJ/m3 but not both).
Content and answers to first round of review:
- justification for considering review papers only as information sources was not provided
- categorization of papers should be simplified and checked once more. Stating sentences line “Articles of 24/131…” (line 648) sounds strange. Check and correct line 748 – “Articles of 21/12…” What is meant by “…9s articles…” (line 861)? Similarly, check sentence (line 540-541): “Categorization into energy (5/23) or process (9/23) engineering or their combination (9/23) is relatively equal.” – I do not get the meaning of the sentence from the text, although I guess what the authors intended by this.
Author Response
Thank you for your feedback and valuable comments. Please find our reactions below.
List of issues (excerpt):
Several formal issues, to be dealt with in subsequent major revision, remained in the manuscript:
- formatting: text still contains some hypertext links causing f.e. reference to chapter 0 and subchapter 0 to appear in the text (line 858); similarly, check the whole Methodology part for correct referring to manuscript parts. Kindly remove any remaining links from the revised manuscript (please read the instructions for authors carefully).
Thank you for your point. In the pdf document downloaded from the MDPI portal after the second revision these links really do not work. However, hypertext links were completely checked and corrected within Revision 1. In the uploaded docx document (in the first revision) all links work correctly as well as in exported pdf document. Even in the docx document downloaded from MDPI portal after the second revision it works. However, in pdf downloaded from the MDPI portal these links do not work. There is probably some mistake in exporting process. We will report this problem to the editor.
- language: language polishing is necessary. Several sentences have a non-standard syntax. Especially the newly added text parts should be revised in this sense + grammar.
Thank you very much for your point. The newly added text really contained syntax and grammar mistakes. These were corrected. As for the remaining (original) text, our article was proofread by a native speaker with an expertise in technical English. We believe that the language quality is now good enough.
- references: further format unification is necessary, see refs. no. 52,18,2 as examples. Decide, whether to state full journal names, or their abbreviations, but do not mix this up.
Thank you for your recommendation. All references were checked and unified.
- line 560: The sentence is to start with a reference to some artwork (probably to Table 5), but the reference is not there.
It is the same issue as the hypertext links above. In our docx and pdf versions (after the first revision) there is no mistake as well as in docx document downloaded from MDPI portal after the second revision.
- Line 652-626: Stating a range of values implies that lower and upper limit are expressed in the same physical units (choose either MJ/kg or MJ/m3 but not both).
Thank you very much for this point. The unit MJ/kg belongs to HHV of pyrolytic oil, while the unit MJ/m3 belongs to LHV of pyrolytic gas. This mistake was corrected by adding the text.
Content and answers to first round of review:
- justification for considering review papers only as information sources was not provided
That is a very good point. The ScienceDirect database offers more than 6000 articles related to rotary kiln research. For the global point of view presented in our study, reviews (about 400) provide much more valuable information. That’s why we chose them for our analysis. Relevant original research papers were included where it was appropriate. The study of the whole extended database took us nearly a year.
- categorization of papers should be simplified and checked once more. Stating sentences line “Articles of 24/131…” (line 648) sounds strange.
It was corrected.
Check and correct line 748 – “Articles of 21/12…”
Thank you very much for your point. The mistake was corrected.
What is meant by “…9s articles…” (line 861)?
Thank you. There was a mistake and it was corrected.
Similarly, check sentence (line 540-541): “Categorization into energy (5/23) or process (9/23) engineering or their combination (9/23) is relatively equal.” – I do not get the meaning of the sentence from the text, although I guess what the authors intended by this.
Thank you for your point. We improve the text in this part to make it clear.
Reviewer 2 Report
Thanks for your explanations and the job done to improve the article.
Author Response
Thank you for your time and valuable feedback.
Round 3
Reviewer 1 Report
Dear authors,
I am glad to report that your manuscript is now fit to proceed to further manuscript processing. My comments were answered to my satisfaction and I believe this review paper will make a worthy contribution in the research field.
A few remaining minor issues can be dealt with at manuscript editing and final proofreading stage. The manuscript I downloaded still shows incorrect and missing links, but I believe it is really some awkward feature of the submission system.
With best regards and wishes for fruitful research continuation,
The reviewer